# GRPO-MA: Multi-Answer Generation in GRPO for Stable and Efficient Chain-of-Thought Training

## Abstract

Recent progress, such as DeepSeek-R1, has shown that the GRPO algorithm, a Reinforcement Learning (RL) approach, can effectively train Chain-of-Thought (CoT) reasoning in Large Language Models (LLMs) and Vision-Language Models (VLMs). In this paper, we analyze three challenges of GRPO: gradient coupling between thoughts and answers, sparse reward signals caused by limited parallel sampling, and unstable advantage estimation. To mitigate these challenges, we propose GRPO-MA, a simple yet theoretically grounded method that leverages multi-answer generation from each thought process, enabling more robust and efficient optimization. Theoretically, we show that the variance of thought advantage decreases as the number of answers per thought increases. Empirically, our gradient analysis confirms this effect, showing that GRPO-MA reduces gradient spikes compared to GRPO. Experiments on math, code, and diverse multimodal tasks demonstrate that GRPO-MA substantially improves performance and training efficiency. Our ablation studies further reveal that increasing the number of answers per thought consistently enhances model performance.

## 1 Introduction

The DeepSeek-R1 model demonstrates that reinforcement learning (RL)—particularly Group Relative Policy Optimization (GRPO) Shao et al. (2024)—is effective for training Chain-of-Thought (CoT) reasoning. GRPO prompts the LLM to generate a reasoning trace before producing the final answer and then reinforces this process via verifiable rewards. Subsequent methods such as DAPO Yu et al. (2025), Dr.GRPO Liu et al. (2025), and GPG Chu et al. (2025) refine GRPO's loss function from different perspectives, achieving more stable training and stronger mathematical reasoning. Beyond text-based tasks, the GRPO paradigm has also expanded to multi-modal domains Chen et al. (2025b); Shen et al. (2025); Huang et al. (2025b); Feng et al. (2025); Song et al. (2025); Kim et al. (2025). These works mainly adopt task-specific reward designs—e.g., temporal video rewards in Video-R1 Feng et al. (2025) or trajectory-distance rewards in ManipVLM-R1 Song et al. (2025)—to improve performance under their respective objectives. Collectively, they show that CoT coupled with verifiable RL rewards substantially enhances multi-modal reasoning. Despite these advances, GRPO still faces several intrinsic limitations that hinder stability, efficiency, and overall effectiveness. These include gradient coupling between thoughts and answers, a sampling–reward trade-off where sparse rewards lead to advantage collapse unless sampling is increased, and unstable advantage estimation.

A well-known issue is the mismatch between reasoning traces and final answers: the reasoning may be valid while the final answer is wrong, or conversely, a flawed reasoning may still yield a correct answer. This phenomenon can be observed in both pure textual reasoning tasks Simoni et al. (2025); Lin et al. (2025a); Paul et al. (2024); Turpin et al. (2023) and multi-modal tasks Chen et al. (2025b); Balasubramanian et al. (2025) including our experiments 5.6. Since the gradients of thoughts and answers are inherently coupled in GRPO, such inconsistencies can distort the gradient direction and consequently undermine training effectiveness. Although GRPO-CARE Chen et al. (2025b) introduces a consistency reward to alleviate this, it risks reward hacking and is difficult to apply when semantic consistency is ill-defined (*e.g.*, it is difficult to judge the consistency between a CoT and the numerical coordinates of a predicted bounding box).

The second challenge concerns the trade-off between reward richness and sampling cost in GRPO. When only a few CoT–answer samples are generated, the reward signal is often too sparse, increasing the likelihood that all rewards in a group are zero and causing advantage collapse, which eliminates meaningful gradients. Increasing the number of samples can reduce this collapse probability, but doing so substantially slows training due to the high cost of generating full CoT–answer pairs. Thus, we need an efficient mechanism that enriches reward signals and reduces advantage-collapse risk while introducing as little sampling overhead as possible.

The third challenge concerns the variance of advantage estimation. From a probabilistic perspective, a "good" thought is one that reliably increases the likelihood of producing a good answer — a property that should be evaluated over the distribution of answers it induces. However, GRPO estimates a thought's advantage from only a single sampled answer, which—especially under high-temperature sampling—introduces substantial variance. More accurate advantage estimates not only reduce training instability but also better guide the model toward internalizing what constitutes a genuinely "good" thought, thereby improving answer quality.

In this paper, we propose **GRPO-MA (GRPO with Multi-Answer)**, a simple yet principled extension of GRPO that arises naturally from examining GRPO-style RL under a unified perspective. For each of $K$ thoughts, we sample $M$ answers. A thought's value is the average reward of its $M$ answers, which is then used to derive its advantage relative to other thoughts, while each of the $K \times M$ answers also receives its own advantage. These two advantages are used to update thought and answer tokens separately. Our theoretical analysis, based on the multivariate delta method Oehlert (1992), shows that $K$ and $M$ play fundamentally different roles in controlling the variance of thought-level advantage estimation. Increasing $M$ monotonically drives the variance toward zero, whereas increasing $K$ only reduces it to a non-zero constant. This provides a theoretical justification for why multi-answer sampling is not only effective but also necessary for stabilizing advantage estimation in GRPO-style algorithms. This design brings three benefits: (1) Averaging rewards across multiple answers reduces gradient coupling from noisy thought–answer mismatches. (2) Sharing $K$ thoughts across $M$ answers is computationally efficient, avoiding the cost of generating $K \times M$ full trajectories while still providing diverse reward signals. (3) Lower-variance value estimates yield more stable advantages and fewer gradient spikes.

We evaluate the effectiveness of GRPO-MA on Code, Math, several distinct vision tasks (Object Detection, Affordance Prediction, Trajectory Prediction, Demand Prediction, OCR-based VQA), and a simulator-based visual manipulation task. Across these diverse domains, GRPO-MA consistently outperforms a GRPO baseline with $K$ responses while adding only marginal training overhead. Compared to a stronger baseline using $K \times M$ responses, GRPO-MA achieves similar or slightly better performance using only about 60% of the training time, highlighting improved sample efficiency from more stable advantage estimation. On the visual manipulation task with extremely sparse rewards, GRPO-MA substantially outperforms standard GRPO. Ablation studies further show that increasing $M$ generally improves performance and that variance reduction in thought-level advantage estimation plays a critical role.

**Contributions.**

- We propose GRPO-MA, a simple but principled improvement over GRPO that is directly motivated by a unified view of challenges in GRPO-style reasoning RL.

- We provide, to the best of our knowledge, the first theoretical variance analysis of chain-of-thought advantage estimation in GRPO-style algorithms, showing that increasing $M$ is necessary for reliably reducing variance.

- Across a wide range of tasks and model configurations, GRPO-MA improves over a GRPO baseline with $K$ responses and slightly exceeds a stronger baseline with $K \times M$ responses using only about 60% of the training time, demonstrating better sample efficiency and greater training stability.

## 2 RELATED WORK

The GRPO algorithm has inspired several works to enhance its stability and efficiency by refining its loss function and sampling strategies. DAPO Yu et al. (2025) introduces several "tricks" to

stabilize training, such as Clip-Higher for exploration, Dynamic Sampling to filter uninformative samples, and a Token-Level Policy Gradient Loss to properly weight complex reasoning chains. Dr. GRPO Liu et al. (2025) corrects inherent response length bias and question difficulty bias by removing specific normalization terms from the loss and advantage calculation, leading to more stable training. Generative Policy Gradient (GPG) Chu et al. (2025) simplifies the GRPO objective and introduces a gradient rescaling method to counteract "zero-gradient" samples, ensuring more effective policy updates. Further research has focused on improving efficiency, with CPPO Lin et al. (2025b) pruning low-impact samples to reduce computational cost and Off-Policy GRPO Mroueh et al. (2025) using stale data to improve sample efficiency. Other works enhance stability, such as GSPO Zheng et al. (2025), which realigns importance sampling at the sequence level; GMPO Zhao et al. (2025), which uses a geometric mean to mitigate sensitivity to outliers ; and GTPO Simoni et al. (2025), which resolves gradient conflicts and prevents policy collapse through trajectory analysis. Additionally, specialized solutions like Spectral Policy Optimization Chen et al. (2025a) create learning signals for "all-negative" sample groups using AI feedback. The work most related to ours is VinePPO Kazemnejad et al. (2024), which performs multi-sample rollouts at every intermediate step in math reasoning. In contrast, we leverage GRPO's structure and apply multi-sampling only at the answer level, producing richer reward signals with substantially less sampling time. We also provide theoretical justification for this design and show its effectiveness across both text reasoning and multimodal tasks.

## 3 PRELIMINARY: GRPO

GRPO Shao et al. (2024) is a PPO-style algorithm Schulman et al. (2017) that computes advantages by normalizing rewards from $K$ sampled responses. For a prompt $p$, GRPO generates responses $\{o_i\}_{i=1}^{K}$ with rewards $\{R_i\}$ and computes $A(o_i) = \frac{R_i - \text{Mean}(\{R_k\})}{\text{Std}(\{R_k\})}$. The complete GRPO objective can be written as:

$$
\mathcal{J}_{\text{GRPO}}(\theta) = \mathop{\mathbb{E}}_{\substack{(p,a)\sim\mathcal{D} \\ o\sim\pi_{\theta_{\text{old}}}}} \left[ \frac{1}{K} \sum_{i=1}^{K} \frac{1}{T_i^o} \sum_{t=1}^{T_i^o} \min(r_t A(o_i), \ \text{clip}(r_t, 1\pm\varepsilon)\, A(o_i)) \right] \\
- \ \beta\, D_{\text{KL}}(\pi_\theta \| \pi_{\text{ref}}),
$$
(1)

where $T_i^o$ denotes the length of the $i$-th output trajectory $o_i = (o_{i,1}, \ldots, o_{i,T_i^o})$, and $\text{clip}(r_t, 1\pm\varepsilon)$ clips the likelihood ratio $r_t$ into the interval $[1 - \varepsilon, \ 1 + \varepsilon]$ to stabilize policy updates. The term $r_t$ is the per-token likelihood ratio, $r_t = \frac{\pi_\theta(o_{i,t}|p,o_{i,<t})}{\pi_{\theta_{\text{old}}}(o_{i,t}|p,o_{i,<t})}$, where $\pi_\theta$ and $\pi_{\theta_{\text{old}}}$ denote the current and behavior policies used for on-policy sampling.

As indicated by Equation 1, the advantage determines whether the probability of a certain token increases or decreases, as well as the magnitude of this change. Therefore, a more stable estimation of the advantage (with lower variance) is beneficial for a more stable model parameter update.

## 4 METHOD

In this section, we first describe the sampling and updating process of GRPO-MA, which builds upon the GRPO framework by introducing a multi-answer sampling strategy. We then analyze the variance change of the advantages via the delta method.

### 4.1 PIPELINE OF GRPO-MA

The core modification in GRPO-MA lies in its sampling pipeline, shown in Fig. 1. Given a prompt $p$, it first generates $K$ thoughts $\{th_1, \ldots, th_K\}$ identically to GRPO. Then, for each thought $th_i$, GRPO-MA then generates $M$ answers $\{ans_{i,1}, \ldots, ans_{i,M}\}$, resulting in $K \times M$ total answers where every $M$ answers share the same thought.

After obtaining rewards $\{R_{i,j}\}_{1 \le i \le K, 1 \le j \le M}$ from a reward function, we define the value of a thought as $V(th_i) = \frac{1}{M} \sum_{j=1}^{M} R_{i,j}$, and normalize it to compute the thought advantage

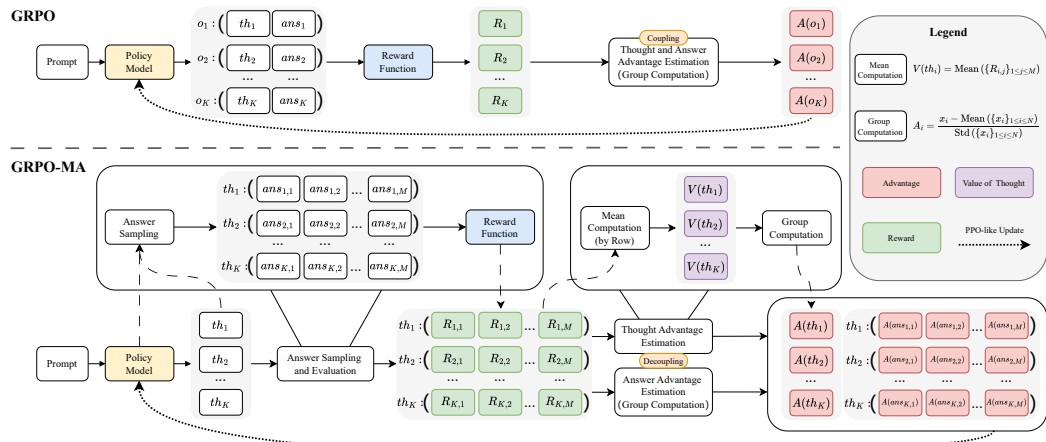

Figure 1: **The operational flow of advantage estimation in GRPO and GRPO-MA**. In the baseline GRPO framework (top), the advantage is computed from a single thought–answer pair, inherently coupling the estimation of thought and answer advantages to a single reward signal. In contrast, GRPO-MA (bottom) extends this setting by sampling multiple answers for each thought. This design decouples the estimation of thought and answer advantages and leverages aggregated information from multiple reward signals, thereby yielding richer supervision and enabling more robust and stable estimation of thought-level advantages.

$A(th_i) = \frac{V(th_i) - \mathrm{Mean}(\{V(th_k)\}_{1 \le k \le K})}{\mathrm{Std}(\{V(th_k)\}_{1 \le k \le K})}$. Similarly, the advantage of an answer is $A(ans_{i,j}) = \frac{R_{i,j} - \mathrm{Mean}(\{R_{k,l}\}_{1 \le k \le K,\, 1 \le l \le M})}{\mathrm{Std}(\{R_{k,l}\}_{1 \le k \le K,\, 1 \le l \le M})}$.

The GRPO-MA objective then combines the two levels of advantages:

$$\mathcal{J}_{\text{GRPO-MA}}(\theta) = \mathop{\mathbb{E}}_{\substack{(p,a)\sim\mathcal{D} \\ \text{th}\sim\pi_{\theta_{\text{old}}}}} \left[ \frac{1}{K} \sum_i \frac{1}{T_i^{\text{th}}} \sum_{t=1}^{T_i^{\text{th}}} \min\left(r_t A(\text{th}_i),\ \mathrm{clip}(r_t, 1\pm\varepsilon)A(\text{th}_i)\right) \right] +$$

$$\mathop{\mathbb{E}}_{\text{ans}\sim\pi_{\theta_{\text{old}}}} \left[ \frac{1}{KM} \sum_{i,j} \frac{1}{T_{i,j}^{\text{ans}}} \sum_{t=1}^{T_{i,j}^{\text{ans}}} \min\left(r_t A(\text{ans}_{i,j}),\ \mathrm{clip}(r_t, 1\pm\varepsilon)A(\text{ans}_{i,j})\right) \right] \tag{2}$$

$$- \beta\, D_{\text{KL}}(\pi_\theta \| \pi_{\text{ref}}),$$

where $T_i^{\text{th}}$ and $T_{i,j}^{\text{ans}}$ denote the lengths of the thought trajectory $\text{th}_i$ and the answer trajectory $\text{ans}_{i,j}$, respectively. The term $A(\text{th}_i)$ is the thought-level advantage computed over the $M$ answers associated with $\text{th}_i$, while $A(\text{ans}_{i,j})$ denotes the advantage of an individual answer. The remaining notation (e.g., $r_t$ and $\mathrm{clip}(r_t, 1\pm\varepsilon)$) follows the same definitions as in the GRPO objective.

## 4.2 VARIANCE OF THE THOUGHT ADVANTAGE

### 4.2.1 PRELIMINARIES

Fix a prompt $p$. Thoughts $th_1, \ldots, th_K$ are sampled independently from a distribution $\pi_\theta(\cdot \mid p)$ conditioned on the prompt $p$.

For each thought $th_i$ we generate $M$ answers independently from the conditional policy, written as $\text{ans}_{i,j} \overset{\text{i.i.d.}}{\sim} \pi_\theta(\cdot \mid p, th_i)$, $j = 1, \ldots, M$. Here the index $j$ simply labels different samples, all drawn from the same distribution $\pi(\cdot \mid p, th_i)$.

Each answer $\text{ans}_{i,j}$ is evaluated by a reward function $r$. We denote the resulting reward as a random variable $R_{i,j} := r(\text{ans}_{i,j}, p)$. $R_{i,j}$ is random because the answer $\text{ans}_{i,j}$ itself is sampled.

Conditioned on a given thought $th_i$ and prompt $p$, the rewards $R_{i,j}$ are i.i.d. with mean $\mu_{R_i}$ and variance $\sigma_{R_i}^2$. The empirical value estimator of thought $th_i$ is the sample mean $V(th_i) = \frac{1}{M} \sum_{j=1}^M R_{i,j}$, with $\mathbb{E}[V(th_i)] = \mu_{R_i}$ and $\text{Var}(V(th_i)) = \frac{\sigma_{R_i}^2}{M}$.

We assume the thought-value estimates $V(th_1), \ldots, V(th_K)$ are independent, *i.e.*, the covariance matrix is diagonal. In practice, small correlations may exist since all thoughts are sampled from the same prompt. However, Appendix A.2.5 shows that the diagonal entries capture most of the covariance energy, suggesting this assumption is largely reasonable.

Finally, define the sample mean and standard deviation across thoughts as $\bar{V} = \frac{1}{K} \sum_{k=1}^K V(th_k)$, $S_V = \sqrt{\frac{1}{K-1} \sum_{k=1}^K (V(th_k) - \bar{V})^2}$, and the thought advantage as $A(th_i) = \frac{V(th_i) - \bar{V}}{S_V}$.

### 4.2.2 Variance of the Thought Advantage

Using the first-order multivariate delta method, and noting that under the independence assumption the covariance matrix of $\{V(th_k)\}$ is diagonal, the variance of the standardized thought advantage is approximated as

$$\text{Var}[A(th_i)] \approx \frac{1}{M \sigma_{\mu_R}^2} \sum_{k=1}^K \left( \delta_{ik} - \frac{1}{K} - \frac{\tilde{\mu}_i \tilde{\mu}_k}{K-1} \right)^2 \sigma_{R_k}^2 \tag{3}$$

where $\delta_{ik}$ is the Kronecker delta ($\delta_{ik} = 1$ if $i = k$, and 0 otherwise), $\mu_{\bar{R}} = \frac{1}{K} \sum_{k=1}^K \mu_{R_k}$ is the average true thought value, $\sigma_{\mu_R}^2 = \frac{1}{K-1} \sum_{k=1}^K (\mu_{R_k} - \mu_{\bar{R}})^2$ is the variance of the true thought values across thoughts, $\tilde{\mu}_i = \frac{\mu_{R_i} - \mu_{\bar{R}}}{\sigma_{\mu_R}}$ is the normalized (expected) advantage of thought $th_i$.

**For the full derivation on the variance of though advantages and answer advantages, please refer to Appendix A.2.**

### 4.2.3 Analysis of the Variance Structure

Combining the above results, we can now compare the effects of increasing $K$ versus increasing $M$.

As $K \to \infty$, by the law of large numbers, the sample variance of the true thought values $\sigma_{\mu_R}^2 = \frac{1}{K-1} \sum_{k=1}^K (\mu_{R_k} - \mu_{\bar{R}})^2$ converges to the population variance $\sigma_\pi^2 = \text{Var}[\mu_{R_i}]$, which characterizes the variability of the true thought values $\mu_{R_i}$ across the population of thoughts sampled from the distribution $\pi_\theta(\cdot \mid p)$. Next, we analyze the sum by splitting it based on the Kronecker delta, $\delta_{ik}$. The single term where $k = i$ converges to a non-zero constant, since expressions like $(\delta_{ik} - \frac{1}{K} - \ldots)$ approach 1. Conversely, the sum of the other $K - 1$ terms, where $k \neq i$, vanishes because each term is of order $O(1/K^2)$, making their total sum $O(1/K)$. This leads to the final result where the variance converges to a limit determined only by the properties of the thought $th_i$ itself:

$$\lim_{K \to \infty} \text{Var}[A(th_i)] \approx \frac{\sigma_{R_i}^2}{M \sigma_\pi^2},$$

In contrast, increasing the number of answers $M$ directly suppresses the variance of the estimated thought value. Since the variance scales inversely with $M$,

$$\text{Var}[A(th_i)] \propto \frac{1}{M},$$

the estimator becomes increasingly accurate, and the total variance *provably approaches zero as* $M \to \infty$.

Taken together, these results reveal a fundamental asymmetry in the variance structure of multi-step reasoning: increasing the number of thoughts $K$ can only reduce variance down to a non-zero floor determined by the inherent variability of the thought population, whereas increasing the number of answers $M$ continues to suppress variance without bound, driving it provably toward zero. This asymmetry highlights the inherent limitation of thought-level sampling and, conversely, establishes answer-level multi-sampling as a principled and quantitatively justified strategy for stabilizing advantage estimation in GRPO-style reasoning algorithms.

Table 1: Task settings and evaluation metrics.

| Task | Input / Output Definition | Evaluation Metric |
|---|---|---|
| **Math** | Input: a math problem. Output: the correct symbolic or numeric solution. | pass@10 / pass@32 Chen et al. (2021b) |
| **Code** | Input: a programming problem. Output: functional solution code. | pass@10 / pass@32 |
| **Object Detection** | Input: an image and a target object name. Output: bounding boxes of the specified object. | Accuracy: proportion of predictions with IoU > threshold. |
| **Affordance Prediction** | Input: an image and a target affordance (e.g., *grasp*, *hold*). Output: 2D affordance coordinates. | Accuracy: proportion of correctly matched points. |
| **Trajectory Prediction** | Input: an image and a manipulation instruction. Output: a 2D end-effector trajectory. | DFD Eiter et al. (1994), HD Huttenlocher et al. (2002), RMSE, EndPoint Dist. |
| **Demand Prediction** | Input: an image and a human demand instruction. Output: the 2D coordinates of the demanded object. | Accuracy: proportion of correct points. |
| **OCR-based VQA** | Input: an image and a text-understanding question (e.g., documents, infographics). Output: an answer string. | ANLS Biten et al. (2019) |

## 5 EXPERIMENTS

We evaluate GRPO-MA on Math Yu et al. (2025), Code PrimeIntellect (2024); White et al. (2024), several distinct vision tasks (Object Detection contributors (2024), Affordance Prediction Myers et al. (2015); Luo et al. (2022), Trajectory Prediction Ji et al. (2025), Demand Prediction Wang et al. (2024), OCR-based VQA Biten et al. (2019); Tito et al. (2021)) and a Simulator-based Visual Manipulation task Li et al. (2024). Our experiments use Qwen2.5-VL-3B-Instruct Bai et al. (2025) as the base model, with all training conducted on four H100 80G GPUs using LoRA Hu et al. (2022) for parameter-efficient fine-tuning. For each task, we conduct a group of experiments separately.

We introduce the $TKAM$ notation to unify the representation of the GRPO and GRPO-MA methods. In this notation, $K$ represents the number of thoughts, and $M$ denotes the number of answers generated per thought. The notation corresponds to GRPO when $M = 1$ and to GRPO-MA when $M > 1$. For instance, T4A4 signifies a process of generating 4 thoughts, with each thought producing 4 answers, resulting in a total of 16 responses.

**More details (datasets, hyperparameters, training settings) are in the appendix A.3 and A.4.**

### 5.1 TEXT AND VISION TASK

#### 5.1.1 TASK SETTING AND METRIC

Table 1 summarizes the input–output format and evaluation metrics for text and vision tasks. These tasks collectively cover a wide range of modalities and reasoning types, enabling a comprehensive evaluation of GRPO-MA.

For **Math**, we adopt a structured output format using `<analysis>`, `<process>`, and `<answer>` tags. GRPO-MA applies multi-sampling on both `<process>` and `<answer>`. For all other tasks, multi-sampling is applied only to `<answer>`.

We track the **Gradient Spike Score (GSS)** Huang et al. (2025a) to measure gradient stability, defined as $\text{GSS}(g_i) = \frac{|g_i|}{\frac{1}{T+1}\sum_{j=0}^{T}|g_j|}$, where $g_j$ represents the gradient at the j-th time step. We report the number of spikes above 10 (GSS@10), where smaller is better. We supplement Appendix A.6.4 with the complete grad_norm and GSS curves for a more intuitive demonstration of training stability. For all tasks, we also report the per-step training time (s) and supplement the wall-clock time

Figure 2: A case study comparing the baseline GRPO with our proposed GRPO-MA on a referring expression grounding task. The prompt is to locate the "purple bottled beverage". The baseline model, GRPO (T4A1), recognizes the target's existence but its reasoning is distracted by other salient objects (the snacks), leading to a failure in grounding. In contrast, our GRPO-MA (T4A4) correctly reasons about the scene's context, focuses on the target object held by the robotic arm, and successfully provides the precise bounding box. This demonstrates the superior robustness of GRPO-MA in complex scene understanding and reasoning.

Table 2: Combined Results for Math and Code Generation Benchmarks. TN: The number of thoughts; AN: The number of answers per thought; S/S: Second/Step during training; Bold indicates the best performance among the GRPO variants.

| Model | TN | AN | Math | | | | Code | | | |
|---|---|---|---|---|---|---|---|---|---|---|
| | | | S/S | GSS | Pass@10 | Pass@32 | S/S | GSS | Pass@10 | Pass@32 |
| Qwen2.5-VL-3B-Ins | | | | | 9.27 | 16.25 | | | 9.80 | 11.67 |
| Qwen2.5-VL-7B-Ins | | | | | 9.97 | 18.39 | | | 10.72 | 11.31 |
| Qwen2.5-VL-72B-Ins | | | | | 33.07 | 41.39 | | | 20.39 | 22.37 |
| SFT | | | | | 11.07 | 18.11 | | | 8.72 | 10.59 |
| GRPO | 4 | 1 | 111.24 | **5** | 11.78 | 20.32 | 76.21 | **6** | 11.56 | 13.70 |
| GRPO | 8 | 1 | 140.05 | 13 | 11.16 | 21.30 | 104.83 | 24 | 11.44 | 13.39 |
| GRPO | 16 | 1 | 225.43 | 15 | 12.89 | 21.72 | 186.91 | 25 | **11.92** | 14.12 |
| GRPO-MA | 4 | 4 | 132.87 | **5** | **14.70** | **27.60** | 93.45 | 10 | 11.69 | **14.70** |

curve to compare the training efficiency of GRPO-MA and the baseline from another perspective in Appendix A.6.3.

### 5.1.2 BASELINES

We adopt models from the Qwen2.5-VL-Instruct series (3B, 7B, and 72B) Bai et al. (2025) as baselines to evaluate the performance of general-purpose models on our tasks. In addition, we train Qwen2.5-VL-3B-Instruct with real labels using supervised fine-tuning (SFT) to compare against GRPO, denoted as **SFT** in the results. Finally, we compare our proposed GRPO-MA with GRPO under different numbers of responses to demonstrate the superiority of GRPO-MA in terms of training efficiency and performance.

### 5.1.3 MAIN RESULTS

The experimental results are presented in Tab. 2 (Math and Code Problem), Tab. 3 (Object Detection, Affordance Prediction and Demand Prediction), and Tab. 4 (OCR-based VQA and Trajectory Prediction). Across multiple visual tasks, our proposed GRPO-MA outperforms both the GRPO and SFT under various settings, demonstrating its excellent versatility across diverse tasks. Compared to T4A1, T4A4 achieves significant performance gains with only about a 15% increase in training time. Compared to T16A1, T4A4 achieves comparable or even slightly better performance with about a 40% reduction in training time, which demonstrates that GRPO-MA does not involve a trade-off between training efficiency and training performance, but rather enhances both simultaneously.

Table 3: Combined Results for Object Detection, Affordance, and Demand Prediction. TN: The number of thoughts; AN: The number of answers per thought; S/S: Second/Step during training; UMD: UMD Part Afforance Dataset; AGD20K: AGD20K Dataset; Bold indicates the best performance among models of the same size.

| Model | TN | AN | Object Detection | | | | | | Affordance Prediction | | | | Demand Prediction | | |
| --- | --- | --- | --- | --- | --- | --- | --- | --- | --- | --- | --- | --- | --- | --- | --- |
| | | | S/S | GSS@10 | IoU@0.5 | IoU@0.6 | IoU@0.7 | IoU@0.8 | S/S | GSS@10 | UMD | AGD20K | S/S | GSS | Accuracy |
| Qwen2.5-VL-3B | | | | | 60.87 | 50.54 | 39.67 | 21.32 | | | 38.98 | 52.73 | | | 11.41 |
| Qwen2.5-VL-7B | | | | | 70.11 | 60.23 | 48.02 | 25.89 | | | 34.65 | 43.29 | | | 19.95 |
| Qwen2.5-VL-72B | | | | | 72.57 | 60.66 | 47.19 | 24.48 | | | 59.13 | 60.59 | | | 26.27 |
| SFT | | | | | 64.63 | 54.73 | 42.43 | 22.96 | | | 66.35 | 53.18 | | | 36.97 |
| GRPO | 4 | 1 | 13.86 | 5 | 65.11 | 56.16 | 43.88 | 23.02 | 14.34 | 11 | 78.91 | 55.60 | 13.74 | 5 | 38.13 |
| GRPO | 8 | 1 | 18.86 | 30 | 67.13 | 57.52 | 42.62 | 22.82 | 18.92 | 14 | 88.14 | 57.90 | 18.20 | 12 | 40.81 |
| GRPO | 16 | 1 | 26.99 | 16 | 69.03 | 60.29 | 45.16 | 24.04 | 24.27 | 22 | 89.32 | 57.24 | 25.42 | 37 | 42.47 |
| GRPO-MA | 4 | 4 | 15.77 | **1** | **69.71** | **61.32** | **46.77** | **25.64** | 15.86 | **5** | **89.96** | **58.40** | 14.33 | 6 | **42.63** |

Table 4: Combined Results for OCR-based VQA and Trajectory Prediction. TN: The number of thoughts; AN: The number of answers per thought; S/S: Second/Step during training. Bold indicates the best performance among models of the same size.

| Model | TN | AN | OCR-based VQA | | | | | Trajectory Prediction | | | | | |
| --- | --- | --- | --- | --- | --- | --- | --- | --- | --- | --- | --- | --- | --- |
| | | | S/S | GSS@10 | Infographics | St VQA | Doc VQA | S/S | GSS@10 | DFD | HD | RMSE | EndPoint |
| Qwen2.5-VL-3B | | | | | 73.10 | 67.63 | 91.33 | | | 571.60 | 537.63 | 404.40 | 429.93 |
| Qwen2.5-VL-7B | | | | | 78.94 | 74.03 | 93.51 | | | 496.44 | 451.19 | 340.33 | 354.03 |
| Qwen2.5-VL-72B | | | | | 79.72 | 74.27 | 93.26 | | | 386.83 | 352.18 | 263.61 | 300.77 |
| SFT | | | | | 74.77 | 69.68 | 92.94 | | | 277.68 | 261.86 | 196.55 | 228.62 |
| GRPO | 4 | 1 | 14.79 | 42 | 73.70 | 68.94 | 93.15 | 29.17 | 14 | 187.99 | 172.58 | 140.80 | 142.74 |
| GRPO | 8 | 1 | 19.62 | 88 | 76.33 | 71.56 | 93.98 | 34.51 | 18 | 172.41 | 157.09 | 130.09 | 137.29 |
| GRPO | 16 | 1 | 26.79 | 68 | 76.65 | 72.25 | 94.20 | 66.55 | 21 | 165.16 | 149.59 | 122.95 | 130.56 |
| GRPO-MA | 4 | 4 | 17.17 | **17** | **76.69** | **72.48** | **94.22** | 35.41 | **10** | **151.10** | **138.29** | **111.59** | **120.60** |

**Gradient Stability** In most experiments, T4A4 achieves the lowest GSS@10, indicating the best gradient stability during training, consistent with our theoretical analysis: as a crucial component of gradient magnitude, the more stable estimation of the advantage value also contributes to greater gradient stability.

**Case Study** As illustrated in Fig. 2, we present a case study to contrast the reasoning processes of T4A4 and T4A1 for the object detection task. T4A4 focuses on the general vicinity of the target object and its surrounding context. Conversely, T4A1 fails to detect the target, instead paying its attention on the central region of the image. Additional case studies are provided in the appendix A.5.

Please see Appendix A.6.6 for more Math and Code dataset evaluation results on GSM8K Cobbe et al. (2021) and HumanEval Chen et al. (2021a).

## 5.2 SIMULATOR-BASED MANIPULATION TASK

### 5.2.1 TASK SETTING

We adapt most of the experimental settings introduced in ManipLLM Li et al. (2024), which provides a simulator-based framework for visual manipulation tasks. To increase the difficulty of the task, we introduce two modifications to the experimental setup. First, to ensure greater observational diversity, the camera is reconfigured to view the target object from a randomly sampled angle in each trial. Second, we adapt a stricter success criterion: an attempt is immediately deemed a failure if the predicted contact point does not lie on the surface of the target object. Following ManipLLM, when the model outputs a grasping point on the image, we execute a rule-based grasping strategy. Specifically, the sucker approaches along the surface normal at the predicted point, and the subsequent trajectory is adjusted depending on the object category. For evaluation, we report the proportion of predicted points that lead to successful manipulation. Through data collection and training, we observe that this task is highly reward-sparse, since solving it requires the model to reason about object-specific interaction dynamics.

### 5.2.2 BASELINES

We adapt some of the same baselines used in visual tasks and added several additional baselines: ManipLLM-7B, CoT-SFT, and GRPO-NoThink.

**ManipLLM-7B** They collects a large number of successful samples in the simulator and constructs multiple task-specific question-answer pairs, utilizing the SFT training approach. We have fine-tuned their weights in the new settings.

**CoT-SFT** We collect successful samples of GRPO-MA-T4A4 (including the chain of thoughts and answers), then fine-tune Qwen2.5-VL-3B using SFT.

**GRPO-NoThink** We employ GRPO to train the Qwen2.5-VL-3B, but we do not require the model to generate a thought process; instead, it directly produces the answers.

### 5.2.3 MAIN RESULTS

The experimental results are presented in Tab. 5. A direct comparison reveals that the performance of T4A4 is significantly superior to that of T4A1. This outcome demonstrates that in tasks with extremely sparse rewards, such as multi-modal manipulation, employing a multi-answer sampling strategy leads to a more stable training process and facilitate sampling of effective responses.

Furthermore, our experiments provide valuable insights into the indispensable role of the Chain of Thought (CoT) in this context. We observe that the GRPO-NoThink model, which ablates the CoT while sampling an equal number of answers as GRPO-MA-T4A4, suffers a substantial degradation in performance. This result, along with the strong performance of the CoT-SFT model, clearly indicates that a high-quality CoT is a critical prerequisite for generating superior answers and effectively tackling such complex tasks.

Table 5: Manipulating Point Prediction. TN: The number of thoughts; AN: The number of answers per thought; S/S: Second/Step during training.

| | | | Success Rate (%) | |
|---|---|---|---|---|
| Model | TN | AN | Seen | Unseen |
| Qwen2.5-VL-3B | | | 4.73% | 1.30% |
| ManipLLM-7B | | | 22.80% | 7.63% |
| SFT | | | 9.17% | 4.28% |
| CoT-SFT | | | 28.18% | 11.79% |
| GRPO | 4 | 1 | 10.75% | 3.94% |
| GRPO-NoThink | 0 | 16 | 10.60% | 2.40% |
| GRPO-MA | 4 | 4 | **31.40%** | **16.00%** |

### 5.3 MORE BASELINE COMPARISON

To present a more comprehensive empirical evaluation, we additionally include three recent GRPO-based variants—GRPO-CARE, DAPO, and Dr.GRPO—on the trajectory prediction task.

We follow each method's original formulation: GRPO-CARE adds a consistency-reward term; DAPO incorporates clip-higher, token-level policy-gradient optimization, and overlong reward shaping; and Dr.GRPO modifies both the advantage computation and the optimization objective. For trajectory prediction, which uses continuous accuracy metrics (range (0,1)), DAPO's Dynamic Sampling module is not applied.

Table 6: Trajectory Prediction with More Baselines.

| Model | DFD | HD | RMSE | EP |
|---|---|---|---|---|
| GRPO | 187.99 | 172.58 | 140.80 | 142.74 |
| GRPO-CARE | 188.23 | 170.82 | 139.83 | 144.85 |
| DAPO | 184.08 | 167.75 | 136.18 | 146.99 |
| Dr.GRPO | 180.56 | 165.95 | 135.86 | 140.05 |
| **GRPO-MA** | **151.10** | **138.29** | **111.59** | **120.60** |

Table 6 shows that all three variants offer modest improvements over vanilla GRPO, while GRPO-MA achieves substantially better performance across all metrics, demonstrating the effectiveness of multi-answer training.

### 5.4 SCALING ANALYSIS

To study how GRPO-MA behaves as model capacity increases, we conduct a scaling experiment on Qwen2.5-VL-7B-Instruct. We compare GRPO-MA with a matched GRPO baseline using identical training settings on the trajectory prediction task . Results are summarized in Table 7.

Table 7: Scaling Results on Trajectory Prediction.

| Model | DFD | HD | RMSE | EP |
|---|---|---|---|---|
| 7B-T8A1 | 167.32 | 152.34 | 129.50 | 134.60 |
| **7B-MA-T8A4** | **134.67** | **121.89** | **103.64** | **103.27** |

Notably, GRPO-MA still outperforms standard GRPO, indicating that the variance-reduction effect and richer reward signals of multi-answer sampling continue to benefit larger models.

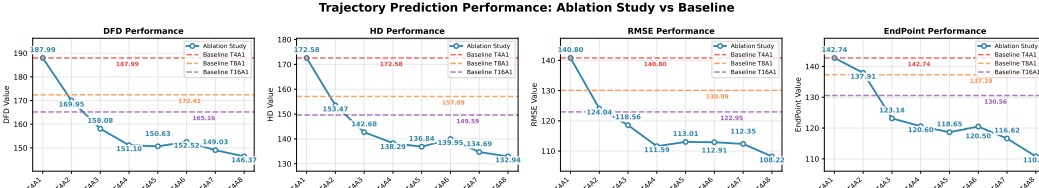

Figure 3: **Ablation Study on Trajectory Prediction** While maintaining the number of thoughts $K = 4$, we gradually increase the number of responses $M$ per thought from 1 to 8 (i.e., the number of responses is 4, 8, 12...32).

## 5.5 ABLATION STUDY

We conduct a detailed ablation study on the trajectory prediction task to analyze the effect of the number of generated answers $M$ per thought, as shown in Fig. 3. The results indicate that as $M$ increases, all evaluation metrics decrease, although the rate of decline becomes progressively smaller.

Surprisingly, T4A3 features 4 thoughts and 12 answers, outperforming T16A1's 16 thoughts and 16 answers across all metrics. One possible explanation for this finding is that the importance of reward signal richness (the number of answers) is less significant than the quality of thoughts; filtering out higher-quality thoughts has a greater impact on the overall training process. Specifically, our method assesses a thought's quality by averaging the rewards of its $M$ subsequent answers ($V(th_i) = \frac{1}{M} \sum_{j=1}^{M} R_{i,j}$). With $M = 3$, T4A3 obtains a more stable and reliable estimate of each thought's value, effectively reducing the noise from any single-answer evaluation. In contrast, T16A1's approach ($M = 1$) is far more susceptible to randomness, as a single, potentially noisy reward is used to judge the entire thought.

## 5.6 INCONSISTENCY ANALYSIS

We quantify the inconsistency between thoughts and answers during training. For a thought $th_i$ with $M$ answers, if $\text{sign}(A(th_i)) \neq \text{sign}(A(ans_{i,j}))$, we mark it as inconsistent. The inconsistency rate is defined as $\text{InconsistencyRate} = \frac{1}{KM} \sum_{i=1}^{K} \sum_{j=1}^{M} \mathbf{1}[A(th_i)A(ans_{i,j}) < 0]$, where $\mathbf{1}[\cdot]$ denotes the indicator function, which equals 1 if the condition inside holds and 0 otherwise.

Under the T4A4 setting, the inconsistency rate is **25.65%** for trajectory prediction and **24.83%** for object detection. Notably, this ratio is also indicative for GRPO baselines (T4A1, T8A1, T16A1), even though they do not explicitly generate multiple answers per thought and thus cannot directly compute it, since they share the same generation hyperparameters (*e.g.*, temperature, top-$k$, and top-$p$ sampling). This observation further supports our claim that inconsistency is common in GRPO's training. Moreover, this inconsistency implicitly undermines model training.

**Accuracy reward curves and richness of reward signal analysis are in the appendix A.6.**

## 6 CONCLUSION

We present GRPO-MA, a simple yet theoretically grounded extension of GRPO that tackles three key challenges in training Chain-of-Thought models: unstable advantage estimation, gradient coupling between thoughts and answers, and sparse reward signals under limited sampling. By generating multiple answers per thought, GRPO-MA reduces the variance of advantage estimation, decouples the gradient between thoughts and answers, and densifies reward feedback. Our theoretical analysis further shows that increasing the number of answers per thought is a principled way to stabilize gradients, which is corroborated by experiments on math, code, and multimodal tasks. Together, these results demonstrate that GRPO-MA improves both the stability and efficiency of GRPO-based reinforcement learning.

**Limitation** Our study has several limitations. First, computational constraints prevent our experiments on larger-scale models. Second, our analysis relies on the simplifying assumption that thought values are independent, a condition that may not hold true in practice. Finally, the lack of a general-purpose reward model means that our testing is confined to tasks with verifiable rewards.

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

# A   APPENDIX

Below is the table of contents for the appendix.

## A.1   MORE RELATED WORK: APPLICATIONS OF GRPO IN MULTIMODAL DOMAINS

VLM-R1 Shen et al. (2025) applies a general GRPO pipeline to Vision-Language Models, enabling smaller models to achieve competitive performance on complex visual reasoning tasks. Vision-R1 Huang et al. (2025b) generates high-quality multimodal Chain-of-Thought data and uses Progressive Thinking Suppression Training (PTST) to prevent the model from creating overly long reasoning paths. Video-R1 Feng et al. (2025) introduces Temporal-GRPO (T-GRPO), a novel reward scheme that encourages the model to leverage temporal information in video sequences. ManipLVM-R1 Song et al. (2025) employs GRPO for robotic manipulation with new affordance-aware and trajectory matching reward functions to improve the localization of interactive parts and the physical plausibility of actions. Robot-R1 Kim et al. (2025) reframes robot learning as a multiple-choice question answering task, using GRPO to optimize the reasoning for embodied manipulation.

## A.2   FULL ANALYSIS OF VARIANCE

This document provides a full derivation of the approximate variance for the Thought Advantage, $A(th_i)$, as presented in the main paper. We first review the multivariate Delta Method, establish the asymptotic normality of our estimators via the Central Limit Theorem (CLT), and finally present the detailed application and gradient calculation.

### A.2.1 THE MULTIVARIATE DELTA METHOD

The Delta Method is a fundamental result in statistics used to approximate the moments of a function of one or more random variables. The multivariate version is central to our analysis.

**General Formulation.** Let $\overrightarrow{V} = (V_1, V_2, \ldots, V_K)$ be a $K$-dimensional random vector of estimators with a true mean vector $\overrightarrow{\mu} = (\mu_1, \mu_2, \ldots, \mu_K)$. Let $M$ denote the sample size used to compute each estimator $V_k$. To emphasize that these estimators are functions of the sample size, we denote the vector as $\overrightarrow{V}_M$. The Delta Method provides the asymptotic distribution of $f(\overrightarrow{V}_M)$ as $M \to \infty$. Specifically, if $\overrightarrow{V}_M$ satisfies the condition for the Central Limit Theorem such that:

$$\sqrt{M}(\overrightarrow{V}_M - \overrightarrow{\mu}) \xrightarrow{d} N(0, \Sigma_{\text{asymptotic}}) \tag{4}$$

where $\xrightarrow{d}$ denotes convergence in distribution, then the transformed variable $f(\overrightarrow{V}_M)$ also converges in distribution:

$$\sqrt{M}(f(\overrightarrow{V}_M) - f(\overrightarrow{\mu})) \xrightarrow{d} N(0, \nabla f(\overrightarrow{\mu})^T \Sigma_{\text{asymptotic}} \nabla f(\overrightarrow{\mu})) \tag{5}$$

From this formal result, we derive the practical formula for approximating the variance of $f(\overrightarrow{V}_M)$ for a large but finite sample size $M$. The term $\sqrt{M}$ acts as a scaling factor that ensures the limiting distribution has a finite, non-zero variance. The variance of the estimator itself is given by:

$$\text{Var}(f(\overrightarrow{V}_M)) \approx \nabla f(\overrightarrow{\mu})^T \text{Var}(\overrightarrow{V}_M) \nabla f(\overrightarrow{\mu}) \tag{6}$$

where $\text{Var}(\overrightarrow{V}_M)$ is the actual covariance matrix of the estimator vector, which is related to the asymptotic covariance by $\text{Var}(\overrightarrow{V}_M) \approx \Sigma_{\text{asymptotic}}/M$.

### A.2.2 ASYMPTOTIC NORMALITY OF THE ESTIMATED VALUE VECTOR

Before applying the Delta Method, we must first establish that our core estimator, the vector of estimated values $\overrightarrow{V(th)}$, satisfies the prerequisite of being asymptotically normal. This justification comes from the Central Limit Theorem (CLT).

For each thought $th_k$, its estimated value $V(th_k)$ is the sample mean of $M$ i.i.d. random variables, the rewards $\{R_{k,j}\}_{j=1}^M$:

$$V(th_k) = \frac{1}{M} \sum_{j=1}^{M} R_{k,j} \tag{7}$$

The rewards have a finite true mean $\mu_{R_k}$ and a finite true variance $\sigma_{R_k}^2$. According to the CLT, as the sample size $M \to \infty$, the distribution of the standardized sample mean converges to a normal distribution. This is formally stated as:

$$\sqrt{M}(V(th_k) - \mu_{R_k}) \xrightarrow{d} N(0, \sigma_{R_k}^2) \tag{8}$$

We now extend this to the full $K$-dimensional vector of estimators, $\overrightarrow{V(th)} = (V(th_1), \ldots, V(th_K))$. Since we have assumed that the estimated values for different thoughts are mutually independent, the joint asymptotic distribution of the vector is also normal. The mean of this limiting distribution is a zero vector, and the covariance matrix is diagonal, composed of the individual variances. Therefore, the entire vector of estimators is asymptotically normal:

$$\sqrt{M}(\overrightarrow{V(th)} - \overrightarrow{\mu}) \xrightarrow{d} N(0, \Sigma_{\text{diag}}) \tag{9}$$

where $\overrightarrow{\mu} = (\mu_{R_1}, \ldots, \mu_{R_K})$ is the vector of true means, and $\Sigma_{\text{diag}}$ is the diagonal covariance matrix of the limiting distribution:

$$\Sigma_{\text{diag}} = \begin{pmatrix} \sigma_{R_1}^2 & 0 & \cdots & 0 \\ 0 & \sigma_{R_2}^2 & \cdots & 0 \\ \vdots & \vdots & \ddots & \vdots \\ 0 & 0 & \cdots & \sigma_{R_K}^2 \end{pmatrix} \tag{10}$$

This result formally justifies the application of the Multivariate Delta Method to the thought advantage function $A(th_i) = f_i(\overrightarrow{V(th)})$.

### A.2.3 Application to the Thought Advantage Function

**Verification of Assumptions.** The prerequisites for the Delta Method are satisfied. First, as established above, our estimator vector $\overrightarrow{V(th)}$ is asymptotically normal. Second, the advantage function $A(th_i) = (V(th_i) - \bar{V})/S_V$ is continuously differentiable everywhere except where the denominator $S_V = 0$, where $\bar{V} = \frac{1}{K} \sum_{k=1}^{K} V(th_k)$ and $S_V = \sqrt{\frac{1}{K-1} \sum_{k=1}^{K} (V(th_k) - \bar{V})^2}$. We evaluate the gradient at $\overrightarrow{\mu}$, where the denominator's analogue is $\sigma_{\mu_R}$. The approximation is thus valid assuming $\sigma_{\mu_R} > 0$, i.e., not all thoughts have the same true value.

**Gradient Calculation.** Let $\overrightarrow{V(th)} = V = (V_1, \ldots, V_K)$ and define

$$f_i(V) = A(th_i) = \frac{N_i(V)}{D(V)} = \frac{V_i - \bar{V}}{S_V}, \qquad \bar{V} = \frac{1}{K} \sum_{j=1}^{K} V_j,$$

$$Q(V) = \frac{1}{K-1} \sum_{j=1}^{K} (V_j - \bar{V})^2, \qquad D(V) = S_V = \sqrt{Q(V)}.$$

We compute $\partial f_i / \partial V_k$ in steps.

First,

$$\frac{\partial \bar{V}}{\partial V_k} = \frac{1}{K}, \qquad \frac{\partial N_i}{\partial V_k} = \delta_{ik} - \frac{1}{K}. \tag{11}$$

For $Q$ we have, using $\partial(V_j - \bar{V})/\partial V_k = \delta_{jk} - \frac{1}{K}$,

$$\frac{\partial Q}{\partial V_k} = \frac{1}{K-1} \sum_{j=1}^{K} 2(V_j - \bar{V})\left(\delta_{jk} - \tfrac{1}{K}\right) \tag{12}$$

$$= \frac{2}{K-1} \left[ (V_k - \bar{V}) - \frac{1}{K} \sum_{j=1}^{K} (V_j - \bar{V}) \right] = \frac{2}{K-1}(V_k - \bar{V}), \tag{13}$$

because $\sum_j (V_j - \bar{V}) = 0$. Therefore

$$\frac{\partial D}{\partial V_k} = \frac{1}{2D} \frac{\partial Q}{\partial V_k} = \frac{V_k - \bar{V}}{(K-1)D}. \tag{14}$$

Applying the quotient rule yields, for arbitrary $V$,

$$\frac{\partial f_i}{\partial V_k} = \frac{(\delta_{ik} - \frac{1}{K})D - (V_i - \bar{V}) \cdot \frac{V_k - \bar{V}}{(K-1)D}}{D^2} = \frac{\delta_{ik} - \frac{1}{K}}{D} - \frac{(V_i - \bar{V})(V_k - \bar{V})}{(K-1)D^3}. \tag{15}$$

Evaluate at $V = \overrightarrow{\mu}$ and denote

$$\sigma_{\mu_R} := D\big|_{V=\mu} = \sqrt{\frac{1}{K-1} \sum_j (\mu_j - \bar{\mu})^2}, \qquad \tilde{\mu}_j := \frac{\mu_j - \bar{\mu}}{\sigma_{\mu_R}}.$$

Then

$$\frac{\partial f_i}{\partial V_k}\bigg|_{V=\mu} = \frac{1}{\sigma_{\mu_R}} \left( \delta_{ik} - \frac{1}{K} - \frac{\tilde{\mu}_i \tilde{\mu}_k}{K-1} \right). \tag{16}$$

Finally, by the first-order multivariate Delta method, with $\text{Var}(\overrightarrow{V}) = \frac{1}{M} \Sigma_{\text{diag}}$ (and $\Sigma_{\text{diag}} = \text{diag}(\sigma_{R_1}^2, \ldots, \sigma_{R_K}^2)$),

$$\text{Var}[A(th_i)] \approx \nabla_V f_i(\mu)^\top \text{Var}(\overrightarrow{V}) \nabla_V f_i(\mu) = \frac{1}{M\sigma_{\mu_R}^2} \sum_{k=1}^{K} \left( \delta_{ik} - \frac{1}{K} - \frac{\tilde{\mu}_i \tilde{\mu}_k}{K-1} \right)^2 \sigma_{R_k}^2. \tag{17}$$

### A.2.4 APPLICATION TO THE ANSWER ADVANTAGE FUNCTION

For a single answer $ans_{i,j}$, the advantage is defined as

$$A(ans_{i,j}) = \frac{R_{i,j} - \overline{R}}{S_R}, \quad \overline{R} = \frac{1}{KM} \sum_{k=1}^{K} \sum_{m=1}^{M} R_{k,m}, \quad S_R = \sqrt{\frac{1}{KM-1} \sum_{k=1}^{K} \sum_{m=1}^{M} (R_{k,m} - \overline{R})^2}.$$
(18)

Using the first-order multivariate Delta method, the variance of $A(ans_{i,j})$ can be approximated as

$$\mathrm{Var}\big[A(ans_{i,j})\big] \approx \nabla_{\mathbf{R}} g_{i,j}(\boldsymbol{\mu})^\top \, \mathrm{diag}(\sigma_{R_1}^2, \dots, \sigma_{R_K}^2, \dots) \, \nabla_{\mathbf{R}} g_{i,j}(\boldsymbol{\mu}),$$
(19)

where $g_{i,j}(\mathbf{R}) = A(ans_{i,j})$ and $\boldsymbol{\mu}$ denotes the vector of reward means.

Evaluating the gradient at $\mathbf{R} = \boldsymbol{\mu}$ and grouping by thought, we obtain

$$\mathrm{Var}\big[A(ans_{i,j})\big] \approx \frac{KM-1}{M(K-1)\sigma_{\mu_R}^2} \sum_{k=1}^{K} \sum_{m=1}^{M} \left( \delta_{(k,m),(i,j)} - \frac{1}{KM} - \frac{\tilde{\mu}_i \tilde{\mu}_k}{M(K-1)} \right)^2 \sigma_{R_k}^2, \quad (20)$$

where $\delta_{(k,m),(i,j)}$ is the Kronecker delta, $\tilde{\mu}_k = (\mu_{R_k} - \mu_{\bar{R}})/\sigma_{\mu_R}$ is the expected advantage of thought $th_k$, $\mu_{\bar{R}} = \frac{1}{K} \sum_{k=1}^{K} \mu_{R_k}$, and $\sigma_{\mu_R}^2 = \frac{1}{K-1} \sum_{k=1}^{K} (\mu_{R_k} - \mu_{\bar{R}})^2$.

### A.2.5 DIAGONALITY ANALYSIS OF MATRICES

To examine whether the assumption of independence across thoughts (i.e., a diagonal covariance matrix) holds in practice, we conducted numerical simulations and empirically estimated the covariance structure of $\overrightarrow{V}(th) = (V_1, \dots, V_K)$. Specifically, we generated $N$ independent replications of the full $K$-dimensional estimator vector, denoted $V^{(n)}$ for $n = 1, \dots, N$, and computed the empirical covariance matrix:

$$\widehat{\Sigma} = \frac{1}{N-1} \sum_{n=1}^{N} \big(V^{(n)} - \overline{V}\big)\big(V^{(n)} - \overline{V}\big)^\top, \qquad \overline{V} = \frac{1}{N} \sum_{n=1}^{N} V^{(n)}.$$
(21)

We then assessed the degree of diagonal dominance using Row-wise strict diagonal dominance and Frobenius-norm based diagonal energy ratio.

**Row-wise strict diagonal dominance.** For each row $i$, the covariance matrix is said to be strictly diagonally dominant if

$$|\widehat{\Sigma}_{ii}| > \sum_{j \neq i} |\widehat{\Sigma}_{ij}|.$$
(22)

We summarize this property by the proportion of rows that satisfy the condition:

$$p_{\mathrm{row\_dom}} = \frac{1}{K} \sum_{i=1}^{K} \mathbf{1}\Big\{ |\widehat{\Sigma}_{ii}| > \sum_{j \neq i} |\widehat{\Sigma}_{ij}| \Big\},$$
(23)

where $\mathbf{1}\{\cdot\}$ denotes the indicator function. A value $p_{\mathrm{row\_dom}} \approx 1$ indicates strong diagonal dominance.

**Frobenius-norm based diagonal energy ratio.** We also consider the proportion of squared Frobenius norm explained by the diagonal entries:

$$\rho_F = \frac{\sum_{i=1}^{K} \widehat{\Sigma}_{ii}^2}{\sum_{i=1}^{K} \sum_{j=1}^{K} \widehat{\Sigma}_{ij}^2}, \qquad 0 \leq \rho_F \leq 1.$$
(24)

Higher values of $\rho_F$ indicate that the diagonal terms dominate the overall covariance energy.

We select 50 samples from the Trajectory Prediction task and, at the 1500-step checkpoint, compute the covariance matrix of the thought-value estimates by performing $N = 10$ independent replications per sample. The empirical results yield $p_{\mathrm{row\_dom}}$=**63.65%** and $\rho_F$ = **70.71%** averaged on 50 samples. Since our theoretical derivations rely on the assumption that the covariance matrix is diagonal, these diagnostics suggest that this assumption has a certain degree of validity in practice, as the estimated covariance matrices exhibit a clear tendency toward diagonal dominance.

## A.3 DETAILS IN TASK SETTINGS

### A.3.1 MATH

We conduct our experiments using problems from the DAPO Yu et al. (2025) training set and evaluate on the AIME2024 test set Maxwell-Jia (2024). The Math training set is constructed by randomly sampling 1,000 problems from the DAPO training corpus. The model is trained for a single epoch on these 1,000 training samples. We do not use a validation set; instead, we select the final model parameters saved at the end of training (the last checkpoint) for testing.

At test time, for each test problem from AIME2024 we generate $n = 100$ independent candidate outputs ("generations"). From these 100 generations we compute the *pass@k* metrics for $k \in \{10, 32\}$.

The reward function is designed with two complementary components: a *format reward* and an *accuracy reward*. The model is required to generate outputs in a predefined structured format:

```
<analysis> xxx </analysis>
<process> xxx </process>
<answer> d </answer>
```

where the answer is represented as a single integes $d$. The format reward assigns a value of 1 if and only if the output strictly follows the required format, and 0 otherwise. The accuracy reward is +1 if the predicted answer is identical to the true answer, and 0 otherwise.

The full prompt template is shown below:

```
{Question} You MUST structure your response using exactly
threesections with XML-style tags in this exact order:
1) <analysis> ... </analysis>
2) <process> ... </process>
3) <answer> ... </answer>

Roles and constraints:
- <analysis>: State relevant concepts, theorems, formulas,
and solution plan. Do NOT perform numeric calculations or
write equations here.
- <process>: Perform ALL detailed computations and
step-by-step derivations based on the analysis. Show
equations and numeric work here.
- <answer>: Output ONLY the final integer (optional sign).
No words, units, punctuation (except the sign), or
explanations.

Hard requirements:
- All three tags must be present and appear in the exact
order <analysis> -> <process> -> <answer>.
- No calculations in <analysis>.
- All computations must be in <process>.
- <answer> must contain a single integer only.
```

**Implementation Note**: In our multi-sample framework, the sampled content encompasses both $<answer>$ and $<process>$ elements.

### A.3.2 CODE

We conduct our experiments using the Python-code portion of the SYNTHETIC-1 dataset PrimeIntellect (2024) and evaluate on the LiveBench code test set White et al. (2024). The Code training set is constructed by randomly sampling 1,000 problems from the SYNTHETIC-1 Python-code corpus. The model is trained for a single epoch on these 1,000 training samples. We do not use a validation set; instead, we select the final model parameters saved at the end of training (the last checkpoint) for testing.

At test time, for each test problem from LiveBench we generate $n = 100$ independent candidate outputs ("generations"). From these 100 generations we compute the *pass@k* metrics for $k \in \{10, 32\}$ as described below.

The reward function is designed with two complementary components: a *format reward* and a *functional (accuracy) reward*. The model is required to generate outputs in a predefined structured format:

```
<think> xxx </think>
<answer> xxx </answer>
```

The format reward assigns a value of $1$ if and only if the output strictly follows the required tag structure and the content within <answer> can be parsed as a syntactically valid Python program. Otherwise the format reward is $0$.

The functional (accuracy) reward is $+1$ if the program inside <answer> executes successfully on the official hidden test inputs, terminates without runtime error, and produces outputs that exactly match the expected outputs for all test cases. Otherwise the accuracy reward is $0$.

The prompt used to condition the model for each problem is exactly:

```
{Question} First output the thinking process
in<think> </think> tags and then output the final code
in <answer> </answer> tags.  The answer should be a
complete Python code solution that solves the given
problem.  Make sure your code handles all edge cases
and follows the input/output format specified in the
problem.  DONOT OUTPUT ANY CODE OR SOLUTION IN THE
THINK TAGS.
```

### A.3.3 Object Detection

We conduct our experiments using the **Agibot World dataset** contributors (2024). The data is partitioned into training, validation, and test sets based on specific `task_ids` from Agibot World dataset. Specifically, the training set is constructed from `task_ids` 424, 480, and 507, comprising a total of 3,000 images (randomly sampling). The validation and test sets are derived from `task_id` 582 and 1352, respectively. For all images, the ground-truth bounding boxes and corresponding object labels are annotated through a crowdsourcing process. The object detection model is trained for a single epoch on the 3,000-image training set.

After training, we perform model selection by evaluating checkpoints on the designated validation set. The model checkpoint that achieves the highest average IoU@0.5 (as defined below) on the validation data is selected for the final evaluation. The performance of this selected model is then reported on the test set.

We evaluate the model's performance using a **IoU rate** metric, which measures the proportion of correctly localized objects based on the Intersection over Union (IoU). A detection is considered positive if the IoU between the predicted bounding box ($B_{pred}$) and the ground-truth bounding box ($B_{gt}$) exceeds a given threshold $\tau$.

The IoU rate at a specific threshold $\tau$, denoted as IoU@$\tau$, is formulated as:

$$\text{IoU@}\tau = \frac{\sum_{i=1}^{N} \mathbf{1}(\text{IoU}(B_{pred}^{(i)}, B_{gt}^{(i)}) > \tau)}{N} \tag{25}$$

where $N$ is the total number of samples in the test set, and $\mathbf{1}(\cdot)$ is the indicator function. To provide a comprehensive assessment, we report the performance across four different IoU thresholds: $\tau \in \{0.5, 0.6, 0.7, 0.8\}$.

The reward function is designed with two complementary components: a *format reward* and an *accuracy reward*. The model is required to generate outputs in a predefined structured format:

```
<think> xxx </think>
<answer> [d, d, d, d] </answer>
```

where the bounding box is represented as a list of four integers $[d, d, d, d]$. The format reward assigns a value of 1 if and only if the output strictly follows the required format, and 0 otherwise. The accuracy reward is defined as the IoU between the predicted bounding box and the ground-truth bounding box.

The full prompt template is shown below:

```
{Question} First output the thinking process in <think>
</think> tags and then output the final answer in <answer>
</answer> tags.  Output the final answer in List format.
Only output the bounding box using [x_min, y_min, x_max,
y_max] format in the final answer.  DO NOT OUTPUT ANY ANSWER
OR CONCLUSION IN THE THINK TAGS.
```

### A.3.4 AFFORDANCE PREDICTION

The task is defined as affordance prediction, where the model, given an image and a specified affordance (e.g., grasping, holding), is required to predict a pixel-wise mask indicating the corresponding region.

We primarily use the UMD Part Affordance Dataset Myers et al. (2015). The official training split of this dataset is used to construct our training and validation sets. Specifically, we use 3,000 images for training and a held-out portion of the original training split for validation. For evaluation, we use the official test split of the UMD dataset. To further assess the model's generalization capabilities, we also use the entire AGD20K dataset Luo et al. (2022) as an additional, challenging test set.

The affordance prediction model is trained for a single epoch on the 3,000-image training set. After training, we perform model selection by evaluating checkpoints on the designated validation set. The model checkpoint that achieves the highest Success Rate (as defined below) on the validation data is selected for the final evaluation. The performance of this selected model is then reported on the test sets (UMD test and AGD20K).

We evaluate the model's performance using a Success Rate metric. This metric measures the proportion of samples where the predicted point correctly falls within the ground-truth affordance mask. A prediction is considered successful if the pixel value at the predicted 2D coordinate is 1 in the ground-truth binary mask.

The Success Rate is formulated as:

$$\text{Success Rate} = \frac{\sum_{i=1}^{N} \mathbf{1}(M_{\text{gt}}^{(i)}(C_{\text{pred}}^{(i)}) = 1)}{N} \tag{26}$$

where $N$ is the total number of samples in the test set, $C_{\text{pred}}^{(i)}$ is the predicted 2D coordinate $(x, y)$ for the $i$-th sample, and $M_{\text{gt}}^{(i)}$ is the corresponding ground-truth affordance mask. The notation $M_{\text{gt}}^{(i)}(C_{\text{pred}}^{(i)})$ represents the value of the mask at the predicted coordinate. $\mathbf{1}(\cdot)$ is the indicator function, which is 1 if the condition is true and 0 otherwise.

The reward function consists of two complementary components: a *format reward* and an *accuracy reward*.

The model is required to generate outputs in the following structured format:

```
<think> xxx </think>
<answer> [d, d] </answer>
```

where the final answer corresponds to a 2D coordinate $[d, d]$, with $d$ denoting an integer. The format reward assigns a value of 1 if and only if the output strictly adheres to this format; otherwise, it is set to 0. The accuracy reward evaluates the correctness of the prediction by checking whether the predicted 2D point lies within the ground-truth affordance mask (i.e., a region where the mask value equals 1). If the prediction falls inside the valid region, +1 reward is given; otherwise, it is not.

The full prompt template is shown below:

```
{Question} First output the thinking process in <think>
</think> tags and then output the final answer in <answer>
```

```
</answer> tags.  Only output one affordance point using [x,
y] format.  DO NOT OUTPUT ANY ANSWER OR CONCLUSION IN THE
THINK TAGS.
```

### A.3.5 TRAJECTORY PREDICTION

The task is defined as trajectory prediction, where the model, given an image and a manipulation instruction, is required to predict the two-dimensional trajectory of the robotic arm's end-effector in the image's pixel coordinate system. The trajectory is represented as a sequence of coordinates, and the predicted path should follow the ground-truth trajectory to successfully complete the instructed manipulation.

We primarily use the trajectory subset of the BAAI ShareRobot dataset Ji et al. (2025). The original dataset is partitioned into training, validation, and test sets. Specifically, we use 3,000 images for training, a held-out portion of the training split for validation, and the test split for evaluation. The model is trained for a single epoch on the 3,000-image training set. After training, we perform model selection by evaluating checkpoints on the designated validation set. The checkpoint that achieves the highest reward value (as defined below) on the validation data is selected for the final evaluation. The performance of this selected model is then reported on the held-out test set.

We evaluate the model's performance using multiple geometric similarity metrics, following the design in ManipVLM-R1 Song et al. (2025). These metrics measure how well the predicted trajectory matches the ground truth from different perspectives. Specifically, we use Discrete Fréchet Distance (DFD), Hausdorff Distance (HD), Root Mean Square Error (RMSE), and Endpoint Distance as evaluation criteria.

The model is required to generate outputs in the following structured format:

```
<think> xxx </think>
<answer> [[x1, y1], [x2, y2], ..., [xn, yn]] </answer>
```

where the final answer corresponds to a variable-length sequence of 2D coordinates $[x, y]$, with $x$ and $y$ denoting integers.

The reward function consists of two complementary components: a *format reward* and a *accuracy reward*. The format reward assigns a value of 1 if and only if the output strictly adheres to this format; otherwise, it is set to 0. To measure how well the predicted trajectory $\hat{T}$ matches the ground-truth trajectory $T^*$, we adopt an accuracy reward following the design in ManipVLM-R1 Song et al. (2025). Specifically, the reward is defined as

$$
\begin{aligned}
R_{\text{acc}} = {} & \exp\big(-k\, D_{\text{DFD}}(\hat{T}, T^*)\big) + \exp\big(-k\, D_{\text{HD}}(\hat{T}, T^*)\big) \\
& + \exp\big(-k\, D_{\text{RMSE}}(\hat{T}, T^*)\big) + \exp\big(-k\, \|\hat{p}_N - p_M^*\|^2\big),
\end{aligned}
\tag{27}
$$

where $D_{\text{DFD}}$, $D_{\text{HD}}$, and $D_{\text{RMSE}}$ denote the Discrete Fréchet Distance, Hausdorff Distance, and Root Mean Square Error between the predicted trajectory $\hat{T}$ and the ground-truth trajectory $T^*$. The final term enforces endpoint accuracy by penalizing the distance between the predicted endpoint $\hat{p}_N$ and the ground-truth endpoint $p_M^*$.

The model is guided by a carefully designed prompt that specifies both the reasoning and the answer requirements. The full prompt template is shown below:

```
{Question} First output the thinking process in <think>
</think> tags and then output the final answer in <answer>
</answer> tags.  Output the final answer in the following
JSON format:  [[x1, y1], [x2, y2], ..., [xn, yn]].  Where
each coordinate pair represents a point in the image's pixel
space and the center of the end effector needs to follow
the coordinates to complete the task.  Each hand trajectory
includes unknown number of [x, y] coordinate pairs.DO NOT
OUTPUT ANY ANSWER OR CONCLUSION IN THE THINK TAGS.
```

A.3.6 DEMAND PREDICTION

The task is defined as demand prediction, where the model, given an image and a human demand instruction (e.g., "I am thirsty"), is required to output a two-dimensional coordinate corresponding to an object in the image that fulfills the demand (e.g., a water bottle or a juice box). A prediction is considered correct if the predicted point lies inside the ground-truth segmentation mask of the demanded object.

We construct the dataset for this task based on MO-DDN Wang et al. (2024), which requires robots to ground a natural demand instruction to objects in the environment. MO-DDN itself is built upon the HSSD scene dataset Khanna et al. (2024), together with a custom demand–object dataset. To build our data, we randomly sample a demand instruction and pair it with a scene containing a target object that satisfies the demand. We then crop and store the corresponding image, resulting in instruction–image pairs.

Following the original MO-DDN splits, we collect data separately from the training and testing tasks. Specifically, we use 3,000 instruction–image pairs as the training set and 1,000 pairs as the validation set, both sampled from the training tasks. For evaluation, we construct a test set of 5,000 instruction–image pairs sampled from thetesting tasks.

We train the model for a single epoch on the training set and perform model selection based on validation accuracy. The checkpoint achieving the highest validation performance is then used for testing, and we report results on the test set.

We evaluate the model's performance using a *Success Rate* metric, defined as the proportion of samples where the predicted coordinate falls within the ground-truth mask of the demanded object. Formally:

$$\text{Success Rate} = \frac{\sum_{i=1}^{N} \mathbf{1}(M_{\text{gt}}^{(i)}(C_{\text{pred}}^{(i)}) = 1)}{N}, \tag{28}$$

where $N$ is the number of samples in the test set, $C_{\text{pred}}^{(i)}$ denotes the predicted 2D coordinate $(x, y)$ for the $i$-th sample, and $M_{\text{gt}}^{(i)}$ is the ground-truth binary mask of the demanded object. The notation $M_{\text{gt}}^{(i)}(C_{\text{pred}}^{(i)})$ indicates the mask value at the predicted location. $\mathbf{1}(\cdot)$ is the indicator function that equals 1 if the condition holds and 0 otherwise.

The reward function for training consists of two complementary components: a *format reward* and an *accuracy reward*. The model must output predictions in the following structured format:

```
<think> xxx </think>
<answer> [d, d] </answer>
```

where the final answer corresponds to a 2D coordinate $[d, d]$, with $d$ denoting an integer. The format reward is assigned 1 if the output strictly follows this structure, and 0 otherwise. The accuracy reward is assigned if and only if the predicted coordinate lies within the ground-truth object mask. These two rewards jointly ensure syntactically valid outputs and semantic correctness.

The model is guided by a prompt template that specifies both the thinking process and the final answer format. The full prompt is given below:

```
You are completing a navigation task where you need to
detect objects from the image that fulfill a user's demand.
The user's demand is {Question}. First output the thinking
process in <think> </think> tags and then output the final
answer in <answer> </answer> tags. Only output one point
using [x, y] format that represents the target demanded
object. DO NOT OUTPUT ANY ANSWER OR CONCLUSION IN THE THINK
TAGS.
```

### A.3.7 OCR-BASED VQA

The task is defined as OCR-based Visual Question Answering (VQA), where the model, given an image containing textual information and a natural language question, is required to output a short natural language answer. The answer must be grounded in the image content and can involve both text extraction and reasoning over visual elements.

We construct the dataset by combining three OCR-based VQA benchmarks: *Document VQA* Tito et al. (2021), *Infographics VQA* Tito et al. (2021), and *Scene Text VQA* Biten et al. (2019). Document VQA focuses on answering questions asked over document images, which may contain printed, typewritten, and handwritten content (*e.g.*, letters, memos, reports). The answers are typically text spans taken verbatim from the document. Infographics VQA considers questions over infographic images containing charts, diagrams, or other structured visual data, where answers are not always explicitly extracted text but can include inferred information. Scene Text VQA consists of natural scene images with embedded text (*e.g.*, storefronts, street signs). The model must jointly leverage OCR reading and visual understanding to answer the questions.

From each of the three training sets, we randomly select 3,000 samples, resulting in a combined training set of 9,000 samples. Additionally, we construct a validation set of 1,500 samples (also drawn from the training splits), while the official validation sets of each benchmark are used as our test set.

The model is trained for a single epoch on the 9,000-sample mixed training set. Model selection is performed based on validation performance, and the checkpoint achieving the highest validation score is reported on the test sets.

The evaluation metric is the *Average Normalized Levenshtein Similarity* (ANLS), which measures the string-level similarity between the predicted and ground-truth answers. ANLS accounts for OCR errors by softly penalizing recognition mistakes. A threshold $\tau = 0.5$ is applied to determine whether a predicted answer is considered valid. Formally, ANLS is defined as:

$$\text{ANLS} = \frac{1}{N} \sum_{i=0}^{N} \left( \max_j s(a_{ij}, o_{q_i}) \right), \tag{29}$$

$$s(a_{ij}, o_{q_i}) = \begin{cases} 1 - NL(a_{ij}, o_{q_i}), & \text{if } NL(a_{ij}, o_{q_i}) < \tau, \\ 0, & \text{if } NL(a_{ij}, o_{q_i}) \geq \tau, \end{cases} \tag{30}$$

where $N$ is the number of questions, $M$ is the number of ground-truth answers per question, $a_{ij}$ is the $j$-th ground-truth answer for the $i$-th question $q_i$, and $o_{q_i}$ is the predicted answer. $NL(\cdot)$ denotes the normalized Levenshtein distance.

The reward function consists of a *format reward* and an *accuracy reward*. The model must output answers in the following structured format:

```
<think> xxx </think>
<answer> xxx </answer>
```

The format reward is 1 if the output strictly follows this structure, and 0 otherwise. The accuracy reward corresponds to the ANLS score of the predicted answer for the current question.

The model is guided by the following prompt template:

```
{Question} First output the thinking process in <think>
</think> tags and then output the final answer in <answer>
</answer> tags.  The answer should be a natural language
text.  The answer should be found in the image.  DO NOT
OUTPUT ANY ANSWER OR CONCLUSION IN THE THINK TAGS.
```

### A.3.8 SIMULATOR-BASED VISUAL MANIPULATION

The task is defined as a simulator-based visual manipulation problem where, given a single RGB observation of a manipulation scene, the model must specify a contact point $(x, y)$ on the object at

which a sucker should attempt to manipulate. The model's output must be grounded in the visual observation and may require reasoning about object geometry, affordances, and reachable contact locations.

We construct the dataset and evaluation splits based on the PartNet Mobility dataset Xiang et al. (2020) and the ManipLLM experimental setup (**A crucial point is that we have followed their setting by using suckers as the end effectors for the robotic arms.**). For training, we adopt the same 20 training categories as ManipLLM, consisting of 1,043 object instances. Training scenes are generated following the SAPIEN simulator Xiang et al. (2020) setup and ManipLLM scene configurations. For testing, we use the open-sourced ManipLLM test set, which contains approximately 1,830 successful test samples spanning both Seen and Unseen objects. To better evaluate model generalization to novel viewpoints, we further construct a camera-perturbed test set by modifying each test sample: the camera orientation vector $[0, 0, 0]$ is replaced by $[x, y, z]$ where each of $x, y, z$ is sampled uniformly from the signed interval $\pm[0.2, 0.6]$. This perturbation preserves other scene properties while intentionally stressing viewpoint robustness. In order to simplify control and isolate contact selection, the sucker approach direction in all experiments is fixed to be the surface normal at the chosen manipulation point $(x, y)$.

The required output must follow a strict format consisting of a reasoning trace and a final contact point, written as:

```
<think> xxxx </think>
<answer> (d,d) </answer>
```

The evaluation metric is Success Rate, following ManipLLM's criterion based on the manipulated object's displacement after the scripted sucker motion. Formally, given $N$ trials,

$$\text{SuccessRate} = \frac{1}{N} \sum_{i=1}^{N} \mathbf{1}\{\text{trial}_i \text{ is successful according to ManipLLM's displacement criterion}\},$$

where $\mathbf{1}\{\cdot\}$ is the indicator function. We report Success Rate on the camera-perturbed test sets, and further provide breakdowns by Seen vs. Unseen objects.

The reward function during GRPO training consists of a format reward and a task reward. The format reward is 1 if the output strictly follows the required structure and 0 otherwise. The task reward is 1 if the manipulation attempt succeeds according to ManipLLM's displacement criterion and 0 otherwise. The overall reward is defined as

$$R_{\text{total}} = R_{\text{format}} + R_{\text{task}},$$

so that only properly formatted and successful outputs receive credit. This ensures that malformed answers cannot be rewarded even if the manipulation itself succeeds.

All experiments are conducted with Qwen2.5-VL-3B as the base model. We train using GRPO for 4,000 optimization steps, selecting checkpoints based on validation success rate. The validation set is constructed by sampling held-out scenes from the same 20 training categories without overlap with the test split. The prompt used in training is as follows:

```
 "system":  "You are an intelligent manipulator.  A
conversation between User and Assistant.  The user
asks a question, and the Assistant solves it.  The
assistant first thinks about the reasoning process
in the mind and then provides the user with the
answer.  The reasoning process and answer are enclosed
within <think> </think> and <answer> </answer> tags,
respectively, i.e.  <think> reasoning process here
</think><answer> answer here </answer>."

"user":  "Specify the contact point (x, y) of
manipulating the object.  The camera resolution
is:'width':  336, 'height':  336, Output
format:  <think>your thinking process</think>
<answer>(x,y)</answer>"
```

Table 8: Hyperparameters for GRPO training.

| Hyperparameter Group | Parameter | Value |
|---|---|---|
| *Training Configuration* | | |
| | Model | Qwen2.5-VL-3B-Instruct |
| | Optimizer | AdamW |
| | Learning Rate ($\eta$) | $1 \times 10^{-5}$ |
| | Batch Size | 1 |
| | Gradient Accumulation Steps | 1 |
| | Total Training Epochs | 1 |
| | Max Completion Length | 4096 |
| | Data Seed | 42 |
| | Floating Point Precision | bfloat16 |
| | Gradient Checkpointing | true |
| | Flash Attention 2 | true |
| *PEFT (LoRA) Configuration* | | |
| | LoRA Rank ($r$) | 64 |
| | LoRA Alpha ($\alpha$) | 128 |
| | LoRA Dropout | 0.05 |
| *GRPO-specific Configuration* | | |
| | Beta ($\beta$) | 0.04 |
| | Epsilon High ($\epsilon_H$) | 0.28 |
| | Epsilon Low ($\epsilon_L$) | 0.2 |
| *Model Specific Configuration* | | |
| | Freeze Vision Modules | true |

## A.4 DETAILS IN TRAINING

### A.4.1 TRAINING HYPERPARAMETERS

We summarize the key hyperparameters used in our GRPO training experiments in Tab. 8. The settings are organized into general, training, and LoRA-related categories for clarity.

### A.4.2 SFT DETAILS

For all Supervised Fine-Tuning (SFT) baselines, we train for 5 epochs. All other settings are kept consistent with GRPO, including the dataset, model selection criteria, and metric calculation.

### A.4.3 GENERATION CONFIGURE

Our model is trained using the Hugging Face `transformers` library (version 4.51.3). During inference, we customize the decoding strategy via the `GenerationConfig` class. Specifically, we set `temperature=1.0` and `do_sample=True` to enable stochastic sampling. We also define `stop_strings=["</think>", "</analysis>"]` only when generating thoughts. The remaining parameters are maintained at their default settings.

## A.5 MORE CASE STUDY AND VISUALIZATION

To provide a more intuitive and in-depth analysis of our model's performance, this section presents a series of curated case studies and visualizations. These examples encompass a range of key tasks, including object detection (Fig. 4) and trajectory prediction (Fig. 5 and Fig. 6). Our aim is to leverage these concrete scenarios to delve into the model's behavior, decision-making logic, and inherent strengths and limitations.

Specifically, in the simulator-based visual manipulation task, we visualize the distribution of the target operation points over multiple sampling attempts in Fig. 7. Green points indicate successful manipulations, while red points represent failures. This visualization demonstrates the robustness of our model.

### A.6 MORE EXPERIMENTAL ANALYSIS

In this section, we further present some experimental results, including the accuracy reward curves during training, and an analysis of the richness of reward signal.

#### A.6.1 ACCURACY REWARD CURVE

We present the accuracy reward curves for five visual tasks in Object Detection (Fig. 8), Affordance Prediction (Fig. 9), Demand Prediction (Fig. 10), OCR-based VQA (Fig. 11) and Trajectory Prediction (Fig. 12). During the curve plotting process, we smooth the curve using a moving average method with a window size of 200. The curves demonstrate that T4A4 (red) exhibits performance comparable to that of T16A1 (blue) in the majority of cases, at times showing a marginal advantage.

#### A.6.2 RICHNESS OF REWARD SIGNALS

For tasks with binary (0-1) rewards, such as Code, Math, Affordance Prediction, Demand Prediction and Simulator-based Visual Manipulation, we compute the proportion of samples whose total reward is positive, which we refer to as the *NoZeroRate*. Formally, it is defined as

$$
\text{NoZeroRate} = \frac{1}{T} \sum_{t=1}^{T} \mathbf{1}\left\{ \left( \sum_{i=1}^{K} \sum_{j=1}^{M} \text{AccR}_{i,j}^{t} \right) > 0 \right\}, \tag{31}
$$

where $\mathbf{1}\{\cdot\}$ denotes the indicator function, which equals 1 if the condition inside holds and 0 otherwise. Here, $T$ is the total number of time steps, $t$ indexes a specific time step, $K$ is the number of thoughts, $M$ is the number of answers per thought, and $\text{AccR}_{i,j}^{t}$ denotes the accuracy reward associated with the $j$-th answer under the $i$-th thought at time step $t$. A higher NoZeroRate indicates a lower proportion of advantage collapses (where collapse means all advantage values become zero), and a higher proportion of effective gradient information contribution.

The statistical results are presented in Tab. 10. We observe that T4A4 achieves the second-highest proportion of non-zero accuracy rewards across all tasks, only behind T16A1. On the one hand, this indicates that under the T4A4 setting, the answers generated by each thought are largely different. On the other hand, it suggests that the diversity of generated answers can be substantially improved by generating additional answers per thought, as shown by the comparison between T4A4 and T4A1.

Table 9: Math and code reasoning results on Qwen3-4B-Instruct.

| Model | Pass@1 (Math) | Pass@1 (Code) |
|---|---|---|
| GRPO-T4A1 | 17.53 | 4.22 |
| GRPO-MA-T4A4 | **18.07** | **6.25** |

#### A.6.3 WALL-CLOCK TIME

We provide wall-clock time to further demonstrate the improvement in training efficiency achieved by the GRPO-MA algorithm. We pick two multimodal tasks and one text reasoning task as examples. The results are shown in Fig. 13, Fig. 14, and Fig. 15. As shown in the figure, GRPO-MA achieves the peak performance of various baselines in a shorter time.

Table 10: NoZeroRate on Different Task. TN: The number of thoughts; AN: The number of answers per thought. **Bold** indicates the best performance and *italics* indicate the second-best performance.

| | TN | AN | Code | Math | Affordance | Demand | Sim Manip |
|---|---|---|---|---|---|---|---|
| GRPO | 4 | 1 | 26.71% | 19.14% | 85.10% | 46.37% | 38.20% |
| GRPO | 8 | 1 | 36.57% | 27.71% | 94.37% | 62.07% | / |
| GRPO | 16 | 1 | **43.71%** | **43.29%** | **97.17%** | **70.20%** | / |
| GRPO-MA | 4 | 4 | *41.86%* | *34.71%* | *96.70%* | *66.47%* | **85.05%** |

Table 11: Additional Results on GSM8K (Math) and HumanEval (Code). TN: Number of thoughts; AN: Number of answers per thought. Bold indicates the best performance among GRPO variants.

| Model | TN | AN | GSM8K (Math) Pass@1 | HumanEval (Code) Pass@1 | Pass@5 |
|---|---|---|---|---|---|
| GRPO | 4 | 1 | 62.54 | 56.10 | 82.93 |
| GRPO | 8 | 1 | 64.67 | 56.83 | 82.93 |
| GRPO | 16 | 1 | 65.13 | 57.80 | 85.37 |
| GRPO-MA | 4 | 4 | **70.58** | **58.66** | **87.20** |

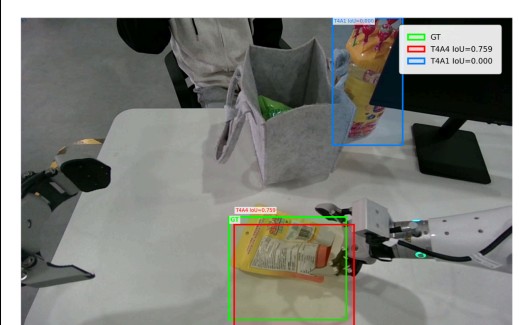

Figure 4: **Case Study on Object Detection** Green text indicates key reasoning content.

### A.6.4 GRAD-NORM AND GSS CURVE

We provide complete Grad Norm and GSS curves, shown in Fig. 16, Fig. 17 and Fig 18. Smaller fluctuations in the Grad Norm curve and lower GSS values indicate fewer gradient spikes during training, resulting in more stable training. The grad norm curve and GSS curve corresponding to GRPO-MA both exhibit smaller fluctuations and GSS values.

### A.6.5 DIFFERENT ARCHITECTURES ANALYSIS

We also evaluate GRPO-MA on a pure-text language model with a different architecture—Qwen3-4B-Instruct—to assess whether our method applies beyond VLM architectures. We use math word problems and programming tasks as representative reasoning benchmarks.

The results are presented in Table 9. GRPO-MA consistently improves Pass@1 accuracy on both math and code reasoning, suggesting that the advantages of multi-answer sampling extend beyond vision–language models and apply to pure-text autoregressive architectures as well.

### A.6.6 MORE RESULTS ON CODE AND MATH

To further examine the generality of our variance-reduction mechanism, we additionally evaluate GRPO-MA on two widely used mathematical and coding benchmarks: GSM8K Cobbe et al. (2021) for math reasoning and HumanEval Chen et al. (2021a) for code generation. These experiments use the same training configuration as our main results, with models trained on the DAPO dataset. For GSM8K, we report the Pass@1 accuracy obtained by running the checkpoint trained for mathematical reasoning. For HumanEval, we report Pass@1 and Pass@5 to assess both direct correctness and sampling-based performance. These findings reinforce our claim that answer-level multi-sampling provides a simple, general, and effective variance-reduction strategy that benefits both mathematical reasoning and code generation tasks. The consolidated results are presented in Table 11.

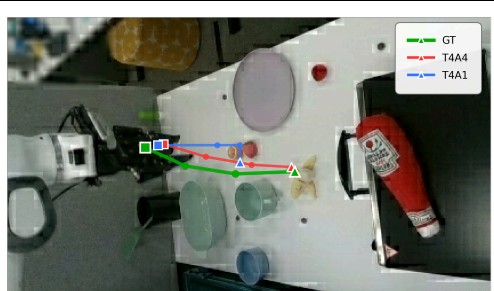

**Prompt:** Record the movement trajectory for completing reach for the strainer

**T4A4:** \<think>\nTo complete the task of reaching for the strainer, the hand should first move towards the left side of the table. The strainer is located on the left side of the table, so the hand should move horizontally to the left. Once the hand is in the general area of the strainer, it should then move vertically downwards to reach the strainer. The final position should be directly above the strainer.\n\</think>\n\<answer>\n[[124, 120], [139, 130], [155, 148], [165, 155], [165, 155]]\n\</answer>

**T4A1:** \<think>\nTo complete the task of reaching for the strainer, the hand should move from its current position to the strainer. The strainer is located on the table, and the hand needs to move towards it. The hand should move horizontally to the right and slightly downward to reach the strainer.\n\</think>\n\<answer>\n[[130, 110], [180, 120], [200, 140], [190, 140]]\n\</answer>

**Discussion:** In T4A4's thought, the target object is first localized relative to the entire tabletop; the arm's approaching direction is then derived from this global pose, and a final fine-tuning command is appended. In contrast, T4A1 merely specifies "move ... to the strainer" without first anchoring it in the scene, so the resulting terminus is biased.

Figure 5: **Case Study on Trajectory Prediction** Green text indicates key reasoning content.

**Prompt:** What are the path coordinates of the robotic arm's end-effector when completing the "reach for the banana" task?

**T4A4:** \<think>\nTo complete the task of reaching for the banana, the mechanical arm needs to move from its current position to the banana. The banana is located towards the right side of the table, slightly above the center. The arm should move horizontally to the right and slightly upwards to reach the banana.\n\</think>\n\<answer>\n[[135, 123], [170, 134], [210, 142], [245, 145]]\n\</answer>

**T4A1:** \<think>\nTo determine the path coordinates for the mechanical arm to reach the banana, we need to consider the initial position of the arm and the target position of the banana. The banana is located on the table, and the mechanical arm needs to move from its current position to the banana's position.\n\nAssuming the table is a flat surface and the banana is placed at a specific location, we can calculate the path coordinates by considering the distance and direction from the current position to the banana's position. If the table is a 2D plane, we can use the following steps to find the coordinates:\n\n1. Identify the current position of the mechanical arm.\n2. Identify the position of the banana on the table.\n3. Calculate the distance and direction from the current position to the banana's position.\n4. Use trigonometry to determine the x and y coordinates of the path.\n\nHowever, without specific pixel coordinates or a visual representation of the table, it's challenging to provide precise coordinates. If we assume the table is a standard 8x8 grid and the banana is located at a specific pixel, we can use the following approach:\n\n- If the banana is at the center of the table, the path would be a straight line from the center to the banana's position.\n- If the banana is at a specific corner, the path would involve moving diagonally or horizontally/vertically to reach the banana.\n\nFor the sake of this example, let's assume the banana is located at the center of the table, which is at (2, 2) on a 10x10 grid (0-indexed).\n\nThe path coordinates from the current position to the banana's position would be:\n[[12, 12], [18, 12], [20, 12], [20, 14]]\n\nThese coordinates represent the path the mechanical arm needs to follow to reach the banana.\n\</think>\n\<answer>\n[[130, 124], [180, 124], [200, 124], [200, 140]]\n\</answer>

**Discussion:** In the T4A4 case, we can clearly observe that the model first identifies the location of the banana, then provides a linguistic description of the end-effector trajectory of the robotic arm, and ultimately produces a trajectory that closely resembles the ground truth. In contrast, in the T4A1 case, although the model continuously proposes schemes for determining a trajectory, it fails to accurately localize the banana. Instead, it keeps making assumptions, which ultimately leads to a significant deviation from the target in the latter part of the trajectory.

Figure 6: **Case Study on Trajectory Prediction** Green text indicates key reasoning content.

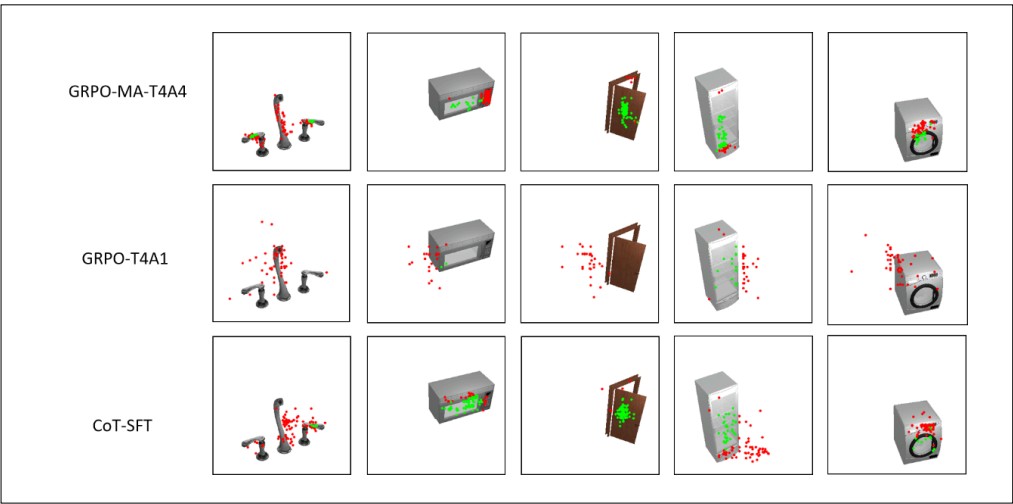

Figure 7: **Visualization on Simulator-based Visual Manipulation** Red dots indicate failures, while green dots represent successes. We can observe that most GRPO-MA-T4A4 points are located on the object. In contrast, GRPO-T4A1 frequently misses the object, resulting in a lower success rate.

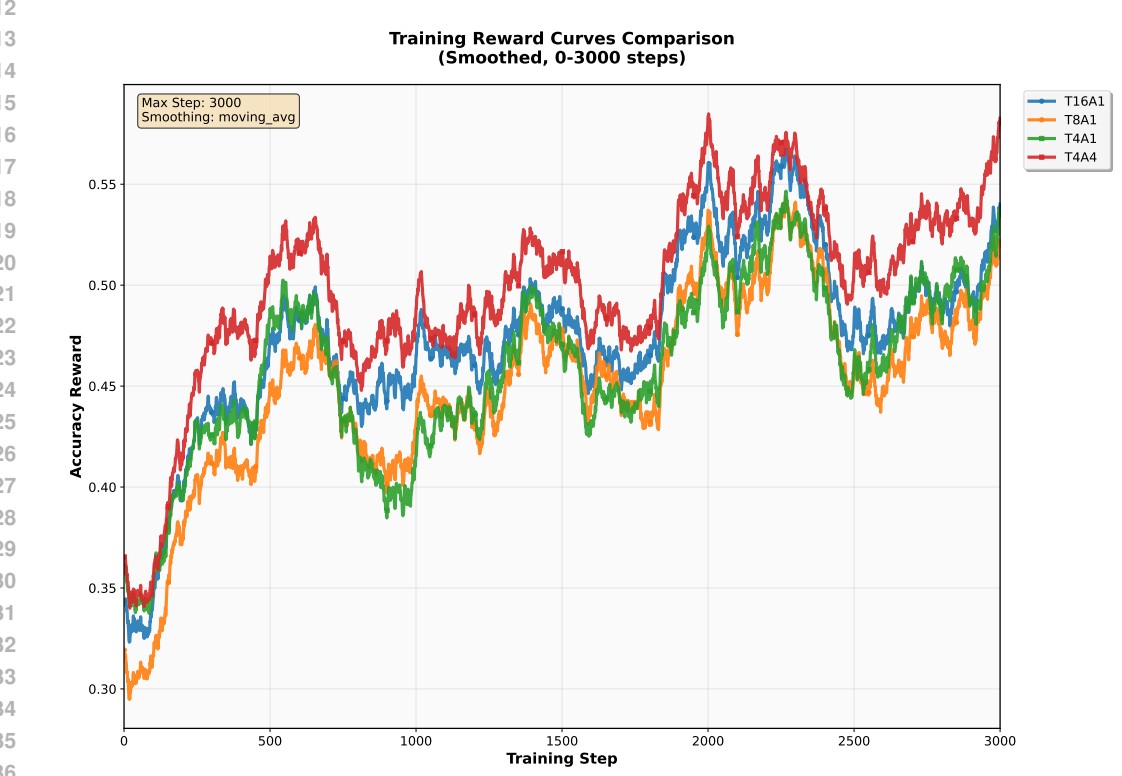

Figure 8: **Accuracy Reward Curve on Object Detection**

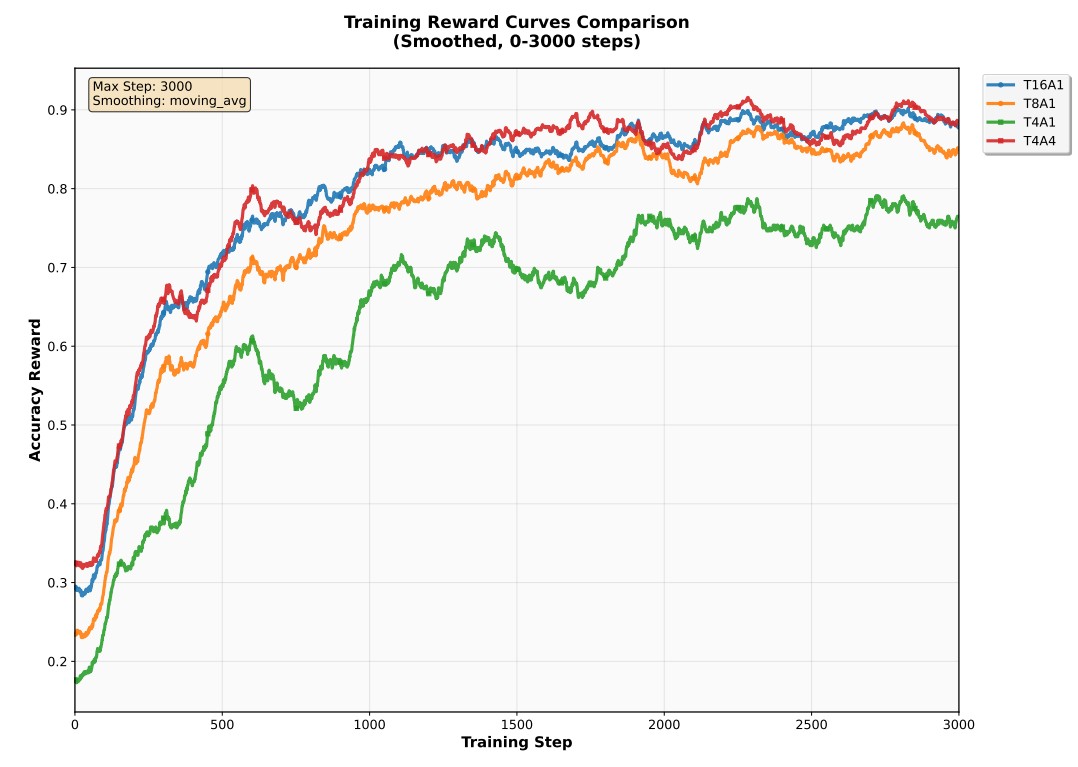

Figure 9: **Accuracy Reward Curve on Affordance Prediction**

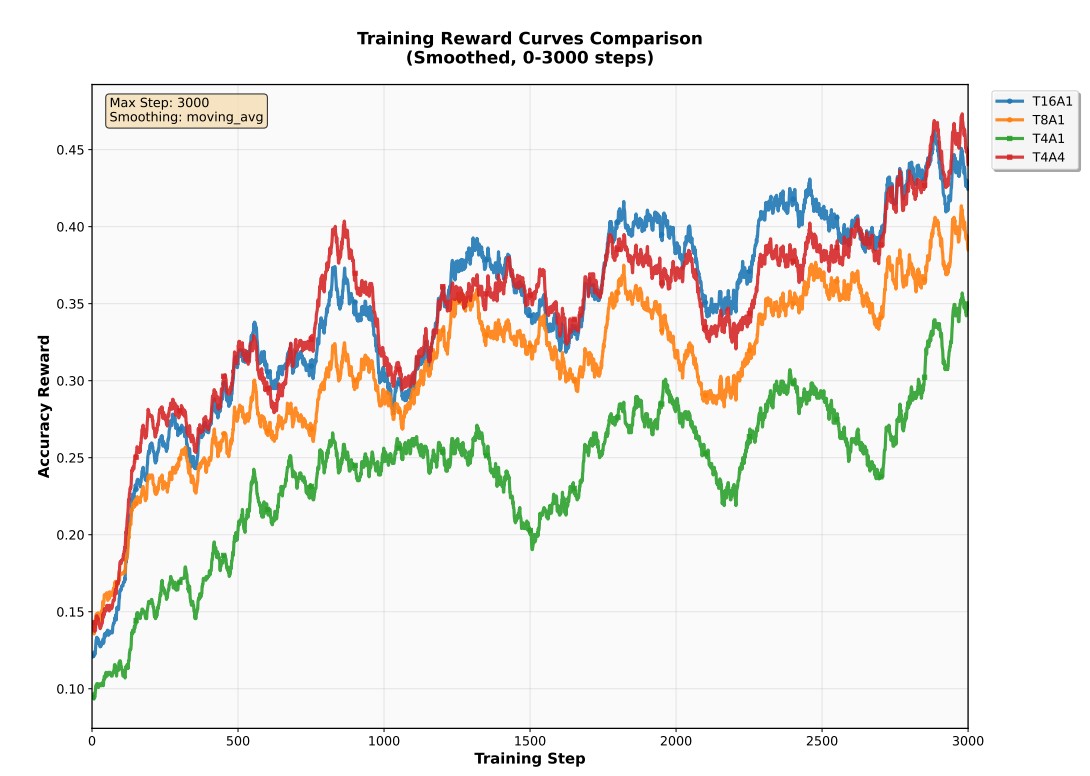

Figure 10: **Accuracy Reward Curve on Demand Prediction**

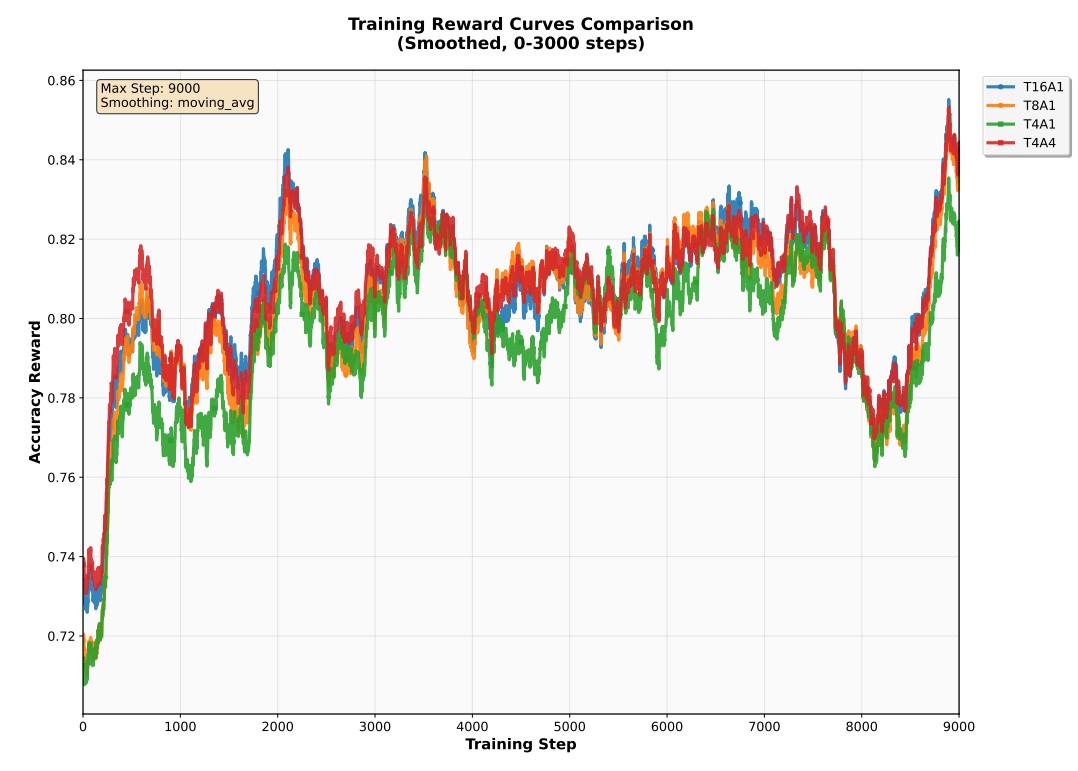

Figure 11: **Accuracy Reward Curve on OCR-based VQA**

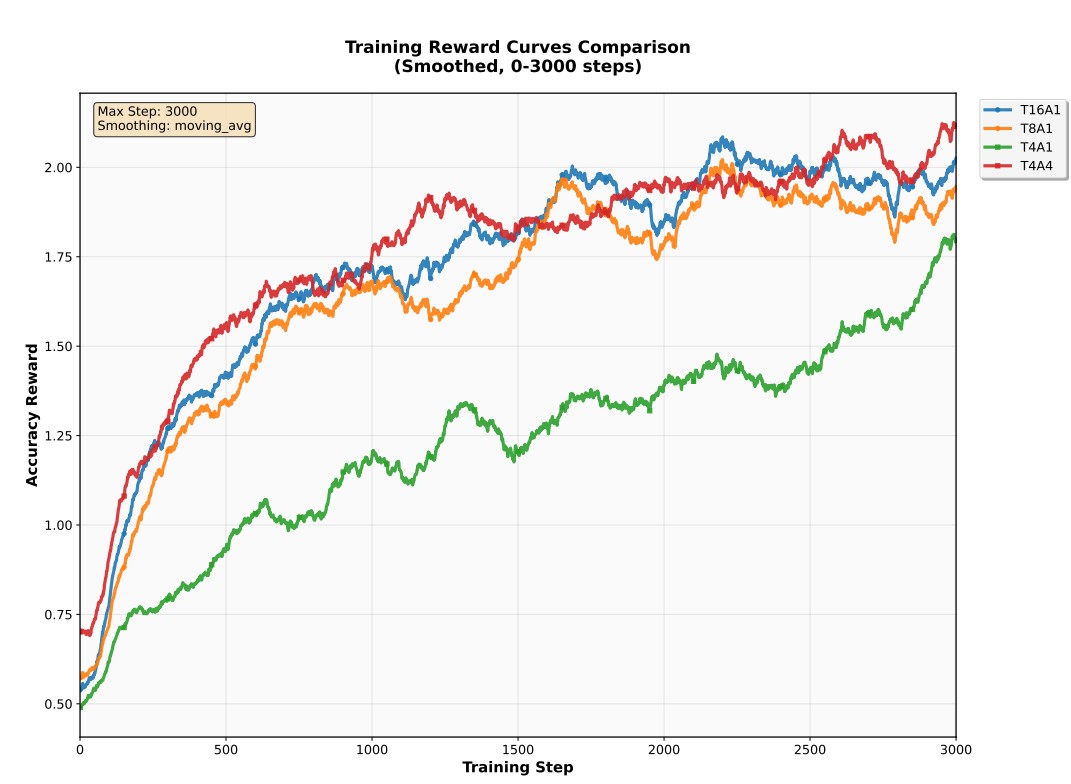

Figure 12: **Accuracy Reward Curve on Trajectory Prediction**

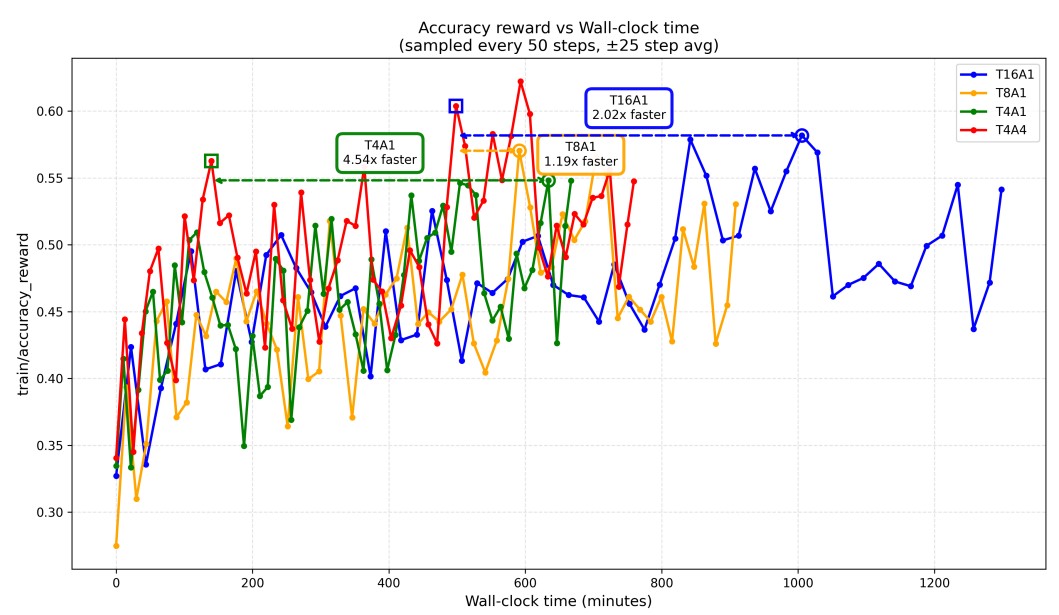

Figure 13: **Wall-clock Time on Object Detection**

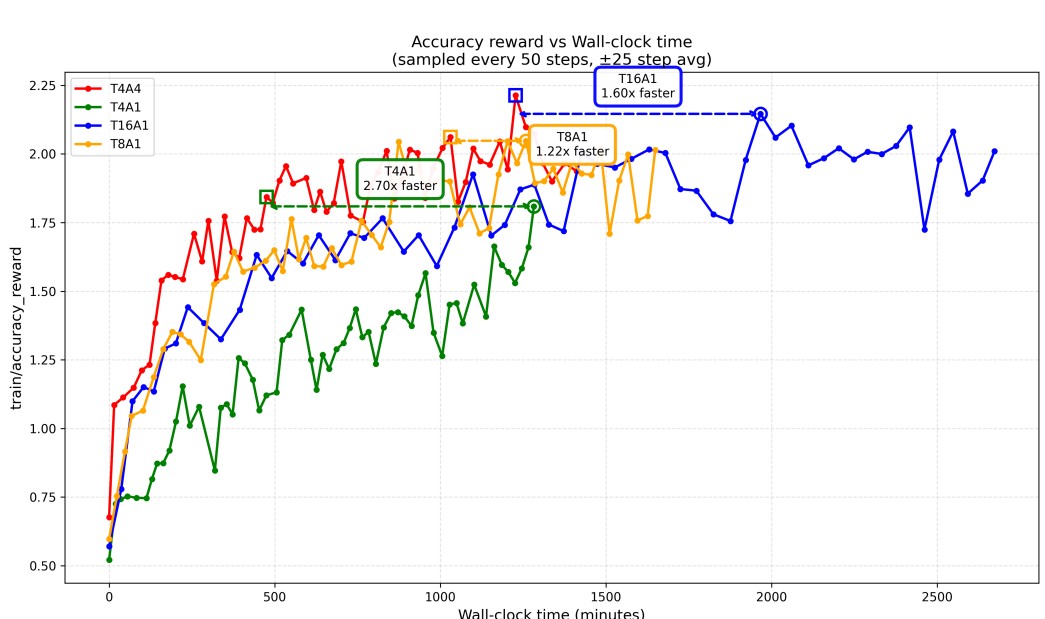

Figure 14: **Wall-clock Time on Trajectory Prediction**

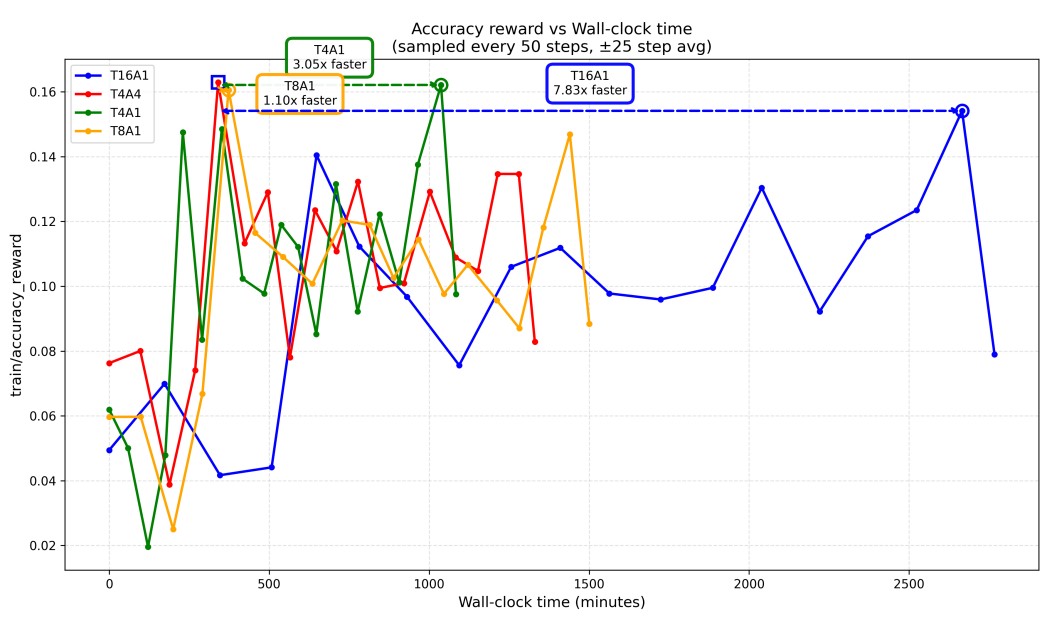

Figure 15: **Wall-clock Time on Code**

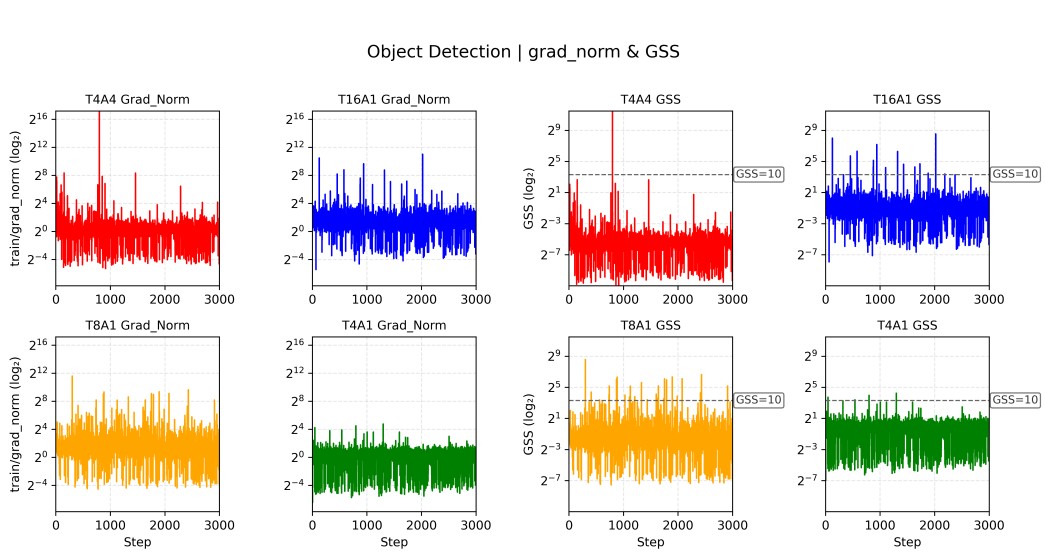

Figure 16: **Grad Norm and GSS on Object Detection**

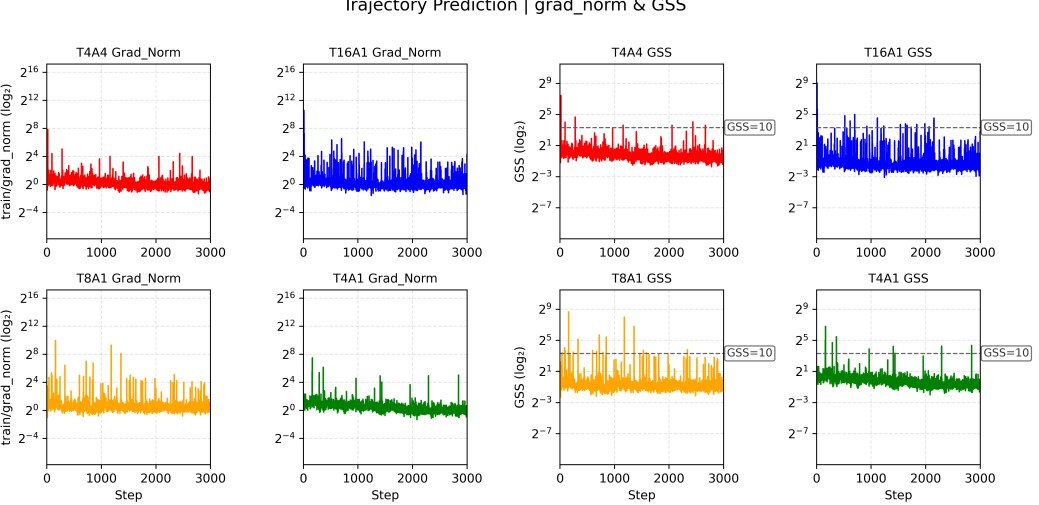

Figure 17: **Grad Norm and GSS on Trajectory Prediction**

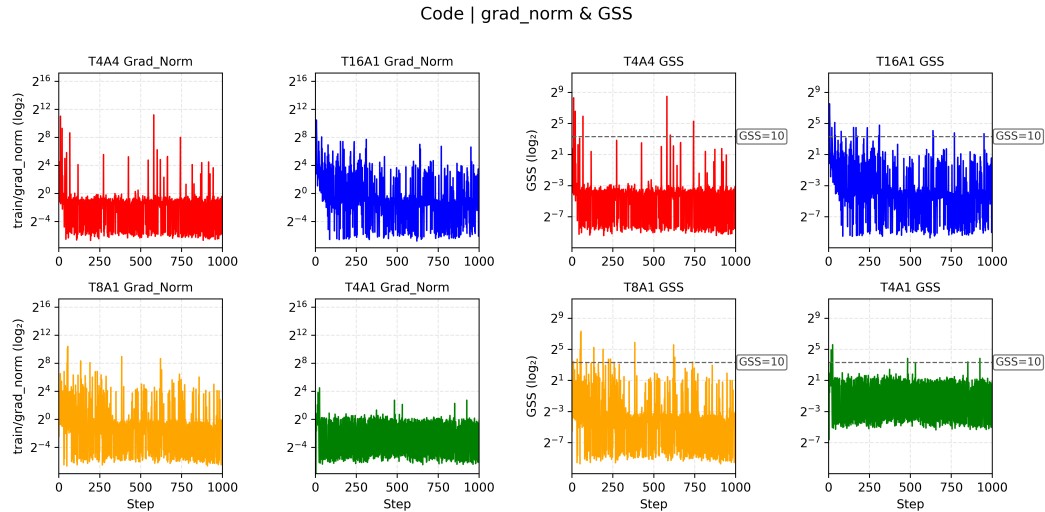

Figure 18: **Grad Norm and GSS on Code**

## A.7 USAGE OF LLMS

We employ a Large Language Model (LLM) to refine the manuscript, with a focus on correcting grammatical errors and enhancing overall readability.

