# OpenReview forum: "GRPO-MA: Multi-Answer Generation in GRPO for Stable and Efficient Chain-of-Thought Training"
_ICLR.cc/2026/Conference — Submitted to ICLR 2026_

### Official Review · Reviewer_5d7r · 2025-10-28

**Soundness:** 2
**Presentation:** 2
**Contribution:** 2
**Rating:** 2
**Confidence:** 4

**Summary:**

This paper proposes GRPO-MA, an extension of the GRPO designed for training CoT reasoning in LLMs and VMs. The method addresses three specific problems in GRPO: gradient coupling between thoughts and answers, sparse reward signals, and unstable advantage estimation. GRPO-MA tackles these by sampling multiple answers per thought, decoupling the learning signals, and aggregating rewards to stabilize training. The paper provides a theoretical analysis of variance reduction with multi-answer sampling ande demonstrates through experiments and ablation studies across mathematical, coding and diverse multimodal tasks that GRPO-MA both improves model performance and enhances training stability and efficiency.

**Strengths:**

1. Clear identification of GRPO's limitations. The paper articulates three concrete challenges with existing GRPO-based RL for CoT—namely, gradient coupling, sparse rewards, and variance in advantage estimation—which are well-motivated in both text and multimodal tasks.
2. Theoretical Contribution. The variance analysis using the delta method is both non-trivial and practically relevant. Equations and accompanying text (see Section 4.2.2, Page 5) give explicit form for the variance of standardized thought advantages, clarifying the distinct influence of increasing the number of thoughts K and answers per thought M.
3. Gradient Stability. The inclusion of the Gradient Spike Score (GSS) is insightful; GRPO-MA consistently achieves lower GSS@10, lending quantitative evidence to theoretical claims of improved training stability (see Table 1 and 2).

**Weaknesses:**

1. Limited Scope of Baseline Comparisons. Although GRPO and SFT are extensively compared, the absense of GRPO-CARE limits the comprehensiveness of the evaluation. Since both GRPO-CARE and the proposed GRPO-MA aim to reduce the inconsistency between reasoning processes (thoughts) and final answers, including GRPO-CARE in the comparison would offer a more direct and compelling demonstration of GRPO-MA’s advantage in mitigating such inconsistencies.
2. Lack of Empirical Analysis on Variance. The paper assumes that the same thought may produce different answers, leading to increased reward variance, which serves as the main motivation for introducing multi-answer sampling. However, the authors do not empirically verify whether a single thought can produce multiple distinct answers to a significant extent in practice, especially under different sampling temperatures. Moreover, although the paper presents a detailed theoretical derivation of variance reduction, the experiments focus primarily on the Gradient Spike Score (GSS@10) as a proxy for training stability. But there is no explanation provided for why 10 is a meaningful cutoff A more direct analysis, such as reporting the step-wise dynamics of gradient norms, would substantially strengthen the empirical support for the claimed stability improvements.
3. Possible Overstatement of Computation Gains. While the paper claims that, for example, T4A4 achieves “comparable or even slightly better performance … with a 40% reduction in training time” vs T16A1 (Page 6), the per-training-step time metric may not reflect overall wall-clock or sample efficiency gains, as actual convergence rates and QA throughput are not shown. It is unclear if there are task or scale-dependent tradeoffs in the number of answers M versus thoughts K, especially for larger models or data regimes.
4. Poor presentation. The paper’s presentation can be improved. The formatting of equations (2) and (3), as well as the description in Section 5.1.1, should be refined. Moreover, a substantial portion of the paper is devoted to explaining GRPO, while the authors should allocate more space to clearly describe their own method and highlight its novel contributions.
5. Lack of Anonymous Repository for Reproducibility. While the paper presents detailed experimental results and comparisons, it does not provide an anonymized code or data repository to support independent replication.

**Questions:**

1. The paper’s approach of drawing multiple samples from the same state and using Monte Carlo estimation to reduce the variance of advantage estimation reminds me of [1]. Could this method be considered a special case of VinePPO, where repeated sampling is applied only at the answer stage? If the authors could clearly explain the similarities and differences between the two methods, it would help make the contribution and novelty of this work clearer.
2. What is the sensitivity of the approach to the independence assumption between thought-value estimates? Could the authors experimentally ablate the effect of correlated thoughts on variance/stability?
3. How does GRPO-MA perform on other challenging mathematical and coding test sets?

[1] VinePPO: Refining Credit Assignment in RL Training of LLMs

---

> ### Author Response · Authors · 2025-11-19
> **Response to Reviewer 5d7r [1/6]**
>
> We sincerely thank you for the exceptionally thorough and constructive review of our work. We are particularly grateful that you found value in our problem identification, theoretical analysis, and evidence for improved training stability.
>
> Your insightful comments regarding the baseline comparisons, empirical variance analysis, and computational efficiency are highly appreciated. In our response below, we have provided additional experimental results and expanded explanations in an effort to engage with your concerns. We hope that these clarifications will help to alleviate your reservations regarding our manuscript.
>
>
>
> ## W1
> > Limited Scope of Baseline Comparisons. Although GRPO and SFT are extensively compared, the absense of GRPO-CARE limits the comprehensiveness of the evaluation. Since both GRPO-CARE and the proposed GRPO-MA aim to reduce the inconsistency between reasoning processes (thoughts) and final answers, including GRPO-CARE in the comparison would offer a more direct and compelling demonstration of GRPO-MA's advantage in mitigating such inconsistencies.
>
> We sincerely appreciate your suggestion, as incorporating additional baselines can indeed enhance the completeness and persuasiveness of our experimental evaluation.
> Following your and other reviewers' suggestions, we have added **Dr.GRPO**, **DAPO**, and **GRPO-CARE** as additional baselines.
> Below, we describe the specific implementation details for each method:
>
> * **GRPO-CARE**: Following the original paper, we added a *consistency reward* term.
> * **DAPO**: We removed the KL-divergence loss and incorporated *clip-higher*, *token-level policy gradient loss*, and *overlong reward shaping*. We did not include *Dynamic Sampling*, since in the trajectory prediction task the accuracy metric is a continuous value in ([0, 1]), rather than a discrete correctness signal.
> * **Dr.GRPO**: In accordance with the original paper, we modified the advantage computation and the optimization objective.
>
> Using these implementations, we report the updated results of all baselines on the trajectory prediction task:
>
> | Model     | DFD           | HD            | RMSE          | EndPoint      |
> |-----------|---------------|---------------|---------------|---------------|
> | GRPO      | 187.99        | 172.58        | 140.80        | 142.74        |
> | GRPO-CARE | 188.23        | 170.82        | 139.83        | 144.85        |
> | DAPO      | 184.08        | 167.75        | 136.18        | 146.99        |
> | Dr.GRPO   | 180.56        | 165.95        | 135.86        | 140.05        |
> | GRPO-MA | **151.10**   | **138.29**    | **111.59**    | **120.60**    |
>
> All experiments above are conducted with 4 thoughts per data sample, and GRPO-MA samples 4 answers for each thought. The experimental results show that multi-answer sampling provides the greatest improvement for GRPO on the trajectory prediction task.
>
> We also wish to clarify that the goal of GRPO-MA is not solely to mitigate thought–answer inconsistency, but also to enrich reward signals and stabilize advantage estimation—two aspects that GRPO-CARE does not explicitly target.
> We hope these extended comparisons help address your concern.

---

> > ### Author Response · Authors · 2025-11-19
> > **Response to Reviewer 5d7r [2/6]**
> >
> > ## W2.1
> > > Lack of Empirical Analysis on Variance. .... However, the authors do not empirically verify whether a single thought can produce multiple distinct answers to a significant extent in practice, especially under different sampling temperatures.
> >
> >
> > We sincerely appreciate your insightful comments regarding the experimental analysis of variance and the motivation behind multi-answer sampling.
> >
> > High-temperature sampling is a widely adopted practice in GRPO-style algorithms because it encourages exploration and yields more diverse generations. Following this standard setting, we use a temperature of 1 in all experiments. Empirically, this temperature consistently produces multiple distinct answers from the same thought, providing the diversity required for effective multi-answer sampling.
> >
> > We apologize that the original draft did not make the empirical evidence sufficiently explicit.
> > **We have, in fact, conducted statistical analyses showing that a single thought can produce multiple distinct answers.** As reported in Appendix A.6.2, we use the NonZeroRate—the proportion of samples for which at least one generated answer is correct—as an indicator of answer diversity. If a single thought were unable to yield diverse answers, the NonZeroRate of GRPO-T4A1 and GRPO-MA-T4A4 would be nearly identical. However, GRPO-MA-T4A4 achieves a noticeably higher NonZeroRate than GRPO-T4A1, confirming that one thought can indeed lead to multiple distinct outputs in practice.
> >
> > Regarding the suggestion to include low-temperature ablations, we fully understand your concern. We note that this behavior arises from the general characteristics of GRPO-style algorithms rather than a design choice specific to our method. GRPO relies on sufficiently diverse, high-temperature samples to maintain exploration and compute meaningful advantages. When the temperature is too low, generations become deterministic, reducing answer diversity and naturally diminishing the benefits of multi-answer sampling.
> >
> > To complement this, we also report a low answer-temperature ablation (T = 0.3), which demonstrates that reduced answer diversity indeed leads to weaker gains from multi-answer sampling—consistent with our theoretical motivation.
> >
> > | Model   | Answer Temperature | DFD        | HD         | RMSE       | EndPoint   |
> > | :------ | :----------------- | :--------- | :--------- | :--------- | :--------- |
> > | GRPO    | 1                  | 187.99     | 172.58     | 140.80     | 142.74     |
> > | GRPO-MA | 0.3                | 176.97     | 164.24     | 131.12     | 138.99     |
> > | GRPO-MA | 1                  | **151.10** | **138.29** | **111.59** | **120.60** |
> >
> > These results further confirm that high-temperature sampling is both a necessary condition for effective multi-answer generation in GRPO-MA and a standard practice in GRPO-based training.
> >
> > ## W2.2
> > > Moreover, although the paper presents a detailed theoretical derivation of variance reduction, the experiments focus primarily on the Gradient Spike Score (GSS@10) as a proxy for training stability. But there is no explanation provided for why 10 is a meaningful cutoff A more direct analysis, such as reporting the step-wise dynamics of gradient norms, would substantially strengthen the empirical support for the claimed stability improvements.
> >
> > Thank you for your valuable suggestions regarding our experimental analysis.
> >
> > Although the choice of 10 is heuristic, our intention was solely to provide a consistent and easy-to-interpret metric. To make the analysis more transparent, we will include gradient-norm curves in Appendix A6.4, alongside the GSS curves. We hope this provides a more transparent view of the training dynamics and more clearly supports our claims regarding stability.

---

> > > ### Author Response · Authors · 2025-11-19
> > > **Response to Reviewer 5d7r [3/6]**
> > >
> > > ## W3.1
> > > > Possible Overstatement of Computation Gains. While the paper claims that, for example, T4A4 achieves “comparable or even slightly better performance … with a 40% reduction in training time” vs T16A1 (Page 6), the per-training-step time metric may not reflect overall wall-clock or sample efficiency gains, as actual convergence rates and QA throughput are not shown.
> > >
> > >
> > > Thank you for pointing out this important issue. Our efficiency comparisons follow the standard practice of fixing the number of optimization steps and sample sizes, as convergence-based reward curves may not provide a stable basis for comparison.
> > >
> > > To complement the per-step metrics, we additionally provide wall-clock training curves in Appendix A6.3. These curves show that GRPO-MA reaches the baseline's peak performance in less time and ultimately achieves higher overall performance.
> > >
> > > *PS: For clarity, we apply an averaging-based smoothing method: every N steps, we compute the average reward over the preceding and following N/2 steps and plot the smoothed values.*
> > >
> > > ## W3.2
> > > >It is unclear if there are task or scale-dependent tradeoffs in the number of answers M versus thoughts K, especially for larger models or data regimes.
> > >
> > >
> > >
> > > Thank you for raising these concerns. We try to address them along three dimensions: task coverage, the roles of K and M, and larger model scales.
> > >
> > > **Task coverage.**
> > >
> > > We evaluate GRPO-MA on two text reasoning tasks, five multimodal tasks, and one simulator-based manipulation task. This set spans diverse modalities and reward structures, and we believe it provides broad empirical coverage for assessing the generality of our method.
> > >
> > > **On K and M.**
> > >
> > > From our theoretical derivation, K and M affect the variance independently. Specifically, increasing M directly reduces the variance of each thought's advantage estimate (since $\mathrm{Var[A(th_i)]}\propto 1/M$), while increasing K introduces more diverse thoughts. This independence allows us to analyze the two hyperparameters separately when determining their appropriate ranges.
> > >
> > > Consistent with this analysis, our theoretical results show that the rate of variance reduction slows as M increases, and our ablations also confirm that when M > 4, the performance gain becomes marginal. In practice, this means that increasing M beyond 4 provides little additional benefit while incurring extra computational cost, which is why we adopt M = 4 as a reasonable and efficient choice across all tasks.
> > >
> > > As larger K generally improves diversity but scales computation linearly, the practical choice of K is primarily constrained by available resources. Under identical training budgets, we found K=4 to provide a strong balance between diversity and compute.
> > >
> > > We believe that generating more diverse thoughts and accurately estimating each thought's advantage both play important roles in effective model optimization.
> > >
> > > **Larger models.**
> > >
> > > We further validate GRPO-MA on a larger model Qwen2.5-VL-7B-Instruct. During training with the 7B model, we used 8 H100 GPUs. Following the guidelines above, we accordingly increased K to 8 while keeping M = 4 unchanged. The results are shown in the table below.
> > >
> > > | Model           | DFD        | HD         | RMSE       | EndPoint   |
> > > | --------------- | ---------- | ---------- | ---------- | ---------- |
> > > | GRPO-7B-T8A1    | 167.32     | 152.34     | 129.50     | 134.60     |
> > > | GRPO-3B-MA-T4A4 | 151.10     | 138.29     | 111.59     | 120.60     |
> > > | GRPO-7B-MA-T8A4 | **134.67** | **121.89** | **103.64** | **103.27** |
> > >
> > > We find that GRPO-MA remains effective at the 7B scale. Unfortunately, extending experiments to larger models is beyond our current computational budget—training the 7B model already requires 8×H100 GPUs with DeepSpeed ZeRO-3 for roughly three days. We sincerely apologize for these limitations. We also note that 7B is a widely adopted model size in recent academic work (e.g., Dr.GRPO, GPG, GRPO-CARE), and we hope these results provide a meaningful reference for evaluating the broader applicability of our method.

---

> > > > ### Author Response · Authors · 2025-11-19
> > > > **Response to Reviewer 5d7r [4/6]**
> > > >
> > > > ## W4
> > > > > Poor presentation. The paper's presentation can be improved. The formatting of equations (2) and (3), as well as the description in Section 5.1.1, should be refined. Moreover, a substantial portion of the paper is devoted to explaining GRPO, while the authors should allocate more space to clearly describe their own method and highlight its novel contributions.
> > > >
> > > >
> > > > Thank you for pointing out these presentation issues. We truly appreciate your suggestions, which helped us improve the clarity of the revision.
> > > >
> > > > We have revised the original Equations (1), (2), and (3) into the new Equations (1) and (2), and we have rewritten both formulations for GRPO and GRPO-MA for improved readability. We have also reformatted the task descriptions in Section 5.1.1 into a clearer table structure and shortened the GRPO algorithm description to better focus on our own contributions.
> > > >
> > > > Thank you again for these valuable suggestions—they have meaningfully improved the overall presentation of the paper.
> > > >
> > > >
> > > >
> > > > ## W5
> > > > > Lack of Anonymous Repository for Reproducibility. While the paper presents detailed experimental results and comparisons, it does not provide an anonymized code or data repository to support independent replication.
> > > >
> > > >
> > > > Thank you for highlighting this important point.
> > > >
> > > > Due to time constraints, practical limitations of our GPU servers, and the internal review procedures at our institution, we are unable to release all code and datasets during the discussion phase. However, we have uploaded to the supplementary material **the complete training code and the trajectory prediction task dataset** at rebuttal period.
> > > >
> > > > We will release the remaining code and configurations as soon as they complete the required approval process. We greatly appreciate your understanding and thank you again for the valuable suggestion.

---

> > > > > ### Author Response · Authors · 2025-11-19
> > > > > **Response to Reviewer 5d7r [5/6]**
> > > > >
> > > > > ## Q1
> > > > > >  the similarities and differences between VinePPO and GRPO-MA
> > > > >
> > > > >
> > > > > We sincerely thank you for pointing out the connection to VinePPO. We found VinePPO to be a highly relevant and insightful work, and we will add a detailed discussion in the related work section.
> > > > >
> > > > > ### **1. Conceptual relation:**
> > > > > At a high level, both VinePPO and GRPO-MA apply repeated sampling from the same underlying state. In this sense, our method is conceptually related to the broader idea introduced by VinePPO. However, GRPO-MA is not a direct instantiation of VinePPO. It is a **GRPO-specific variant** designed around the structure of GRPO-style CoT training and the fact that GRPO's reward is defined solely on the final answer.
> > > > >
> > > > > ### **2. Sampling design: GRPO-specific answer-level multi-sampling:**
> > > > > VinePPO proposes a general framework that allows multi-sampling at multiple intermediate reasoning steps, which requires manually specifying a segmentation of the response (e.g., by sentences or formulas in math reasoning) for different math datasets. In contrast, GRPO-MA performs multi-sampling **only at the answer level**, which is a natural choice in GRPO because the accuracy reward is defined solely on the final answer token sequence.
> > > > >
> > > > > GRPO-MA requires only a clear definition of what constitutes an "answer" for each task, without the need to manually segment the entire reasoning trajectory.
> > > > > For example, in math tasks, the answer naturally includes both the calculation steps and the final numeric result; in multimodal tasks, the answer may instead be a bounding box, a 2D point, or another task-specific structured output. This makes answer-level sampling more intuitive and easier to apply consistently across diverse domains.
> > > > >
> > > > > From this perspective, GRPO-MA can be viewed as a **simplified, GRPO-tailored realization** of the repeated-sampling idea, designed specifically for answer-based reward structures.
> > > > >
> > > > > ### **3. Theoretical contribution: variance decomposition specific to GRPO:**
> > > > > While VinePPO provides strong empirical evidence that multi-sampling improves stability, our work focuses on a **GRPO-specific variance analysis**. In particular, we derive closed-form expressions for the variance of standardized thought advantages under GRPO, which show that:
> > > > >
> > > > > * Increasing the number of thoughts (K) only reduces variance down to a non-zero floor. This implies that it is impossible to precisely estimate the advantage value of thought by increasing K.
> > > > > * Increasing the number of answers (M) drives the variance of the advantage estimate toward zero (under mild independence assumptions), providing a principled justification for prioritizing answer-level multi-sampling in GRPO.
> > > > >
> > > > > To the best of our knowledge, this **explicit decomposition of how (K) and (M) affect variance in GRPO** is not present in prior work. In addition, the variance decomposition we derive is not only specific to GRPO but also offers complementary theoretical support for VinePPO. Since VinePPO is likewise built on the idea of repeated sampling from the same state, our analysis clarifies why repeated sampling systematically reduces estimation variance and thus provides a more solid theoretical justification for the general principle underlying VinePPO's design.
> > > > >
> > > > > ### **4. Reward-signal perspective:**
> > > > > In GRPO, the reward function is typically sparse and attached only to the final answer (e.g., exact match, task-specific score). In this setting, multi-answer sampling is not only a variance-reduction technique but also an efficient way to **densify and enrich the reward signal**: multiple answers for the same thought provide multiple reward observations for the same underlying reasoning trajectory, without requiring additional thought generation. We analyze this effect in more detail in Appendix A6.2.
> > > > >
> > > > > ### **5. Empirical scope:**
> > > > > Finally, we evaluate GRPO-MA across **pure text reasoning tasks, multimodal understanding tasks, and a simulator-based manipulation task**. This empirical scope is intentionally chosen to reflect the broader use cases of GRPO-style training beyond math/coding, and to demonstrate that answer-level multi-sampling remains effective in settings with heterogeneous reward structures and modalities.
> > > > >
> > > > > Overall, we view GRPO-MA as **complementary** to VinePPO: VinePPO introduces a powerful and general multi-sampling scheme, while our work develops a GRPO-specific variant that (i) focuses on answer-level sampling aligned with GRPO's reward structure, (ii) provides a variance decomposition that separates the roles of (K) and (M), and (iii) validates the approach across a wider range of GRPO-style applications, including multimodal and simulator-based tasks.
> > > > >
> > > > > We would like to once again express our sincere appreciation to the reviewer for highlighting VinePPO. It is an excellent and influential contribution to the field, and we greatly value the insights it brings to understanding multi-sampling in policy optimization.

---

> > > > > > ### Author Response · Authors · 2025-11-19
> > > > > > **Response to Reviewer 5d7r [6/6]**
> > > > > >
> > > > > > ## Q2
> > > > > > > What is the sensitivity of the approach to the independence assumption between thought-value estimates? Could the authors experimentally ablate the effect of correlated thoughts on variance/stability?
> > > > > >
> > > > > >
> > > > > > Thank you for raising this important question. We would like to emphasize that the correlation structure among thoughts is not something we can directly control as a free hyperparameter. Once the model architecture and decoding procedure are fixed, the dependence between thought-value estimates is primarily induced by the model’s own generative process, rather than by an externally controllable noise source. In this sense, it is difficult to construct experiments that exactly enforce a prescribed level of correlation between thoughts in a fully controlled manner.
> > > > > >
> > > > > > As shown in Appendix A2.5, under our default hyperparameters, the estimated covariance matrices of thought-value estimates exhibit strong diagonal dominance. This provides indirect but meaningful evidence that sampled thoughts behave approximately independently in practice.
> > > > > >
> > > > > > To further examine the sensitivity of this assumption, we conduct an additional analysis by varying the sampling temperature at the thought level while keeping all other experimental settings identical to Appendix A2.5. This setup isolates the effect of temperature on the correlation structure, as all model configurations, data, and covariance-estimation procedures remain fixed. The results are summarized below:
> > > > > >
> > > > > > | Temperature | $p_{\mathrm{row_dom}}$ | $\rho_{F}$ |
> > > > > > | ----------- | ---------------------- | ---------- |
> > > > > > | 0.5         | 49.48%                 | 53.83%     |
> > > > > > | 1.0         | 63.65%                 | 70.71%     |
> > > > > >
> > > > > > These results demonstrate a clear trend: lower sampling temperatures reduce output diversity and induce stronger correlations among thoughts, whereas higher temperatures increase entropy and lead to more independent thought samples. This aligns with the expected behavior of autoregressive models and confirms that sampling temperature systematically affects the dependence structure among generated thoughts.
> > > > > >
> > > > > > Taken together, these empirical observations support the use of relatively high sampling temperatures during training, as they help maintain the approximate independence assumed by our theoretical analysis. We also note that high-temperature sampling is the standard practice in GRPO-style reasoning frameworks more broadly, as it is essential for maintaining sufficient output diversity and enabling meaningful advantage estimation.
> > > > > >
> > > > > > ## Q3
> > > > > > > How does GRPO-MA perform on other challenging mathematical and coding test sets?
> > > > > >
> > > > > >
> > > > > > Thank you very much for your valuable suggestion regarding additional experiments.
> > > > > > We have supplemented the test results on GSM8K using the checkpoint trained on the DAPO dataset. Unfortunately, due to time constraints, we are unable to provide results on other coding datasets at this stage. We sincerely apologize for this limitation.
> > > > > >
> > > > > > | Model   | TN | AN | Pass@1 on GSM8K |
> > > > > > | ------- | -- | -- | --------------- |
> > > > > > | GRPO    | 4  | 1  | 62.54           |
> > > > > > | GRPO    | 8  | 1  | 64.67           |
> > > > > > | GRPO    | 16 | 1  | 65.13           |
> > > > > > | GRPO-MA | 4  | 4  | **70.58**       |
> > > > > >
> > > > > > From these results, we observe that GRPO-MA continues to demonstrate superior performance on the GSM8K test set.

---

> > > > > > ### Comment · Reviewer_5d7r · 2025-11-24
> > > > > >
> > > > > > Thanks for your detailed explanation and add more experiments to address my concerns. I have increased my score！

---

> > > > > > > ### Author Response · Authors · 2025-11-24
> > > > > > >
> > > > > > > Thank you very much for your thoughtful follow-up and for taking the time to re-evaluate our response. We truly appreciate your positive feedback and are glad that our clarifications and additional experiments helped address your concerns.
> > > > > > >
> > > > > > > If any remaining point would benefit from further clarification, we would be happy to provide additional details.
> > > > > > >
> > > > > > > Thank you again for your constructive comments and for your engagement with our work. We sincerely appreciate it.

---

> > > > > > > ### Author Response · Authors · 2025-11-25
> > > > > > > **GRPO-MA on HumanEval Code Benchmark**
> > > > > > >
> > > > > > > ## Q3
> > > > > > > > How does GRPO-MA perform on other challenging mathematical and coding test sets?
> > > > > > >
> > > > > > > Thank you very much for taking the time to review the newly added experimental results and for your helpful suggestion regarding broader evaluation on challenging mathematical and coding datasets. We have now supplemented the results on the HumanEval benchmark[1] in addition to the GSM8K results provided earlier.
> > > > > > >
> > > > > > > | Model   | TN | AN | Pass@1 on HumanEval |Pass@5 on HumanEval |
> > > > > > > | ------- | -- | -- | --------------- | --------------- |
> > > > > > > | GRPO    | 4  | 1  | 56.10           | 82.93           |
> > > > > > > | GRPO    | 8  | 1  | 56.83           | 82.93           |
> > > > > > > | GRPO    | 16 | 1  | 57.80            |85.37            |
> > > > > > > | GRPO-MA | 4  | 4  | **58.66**            | **87.20**            |
> > > > > > >
> > > > > > > Across both GSM8K and HumanEval, we consistently observe that GRPO-MA improves over GRPO under comparable compute budgets. These results further support our claim that answer-level multi-sampling provides a general and effective variance-reduction strategy that benefits both mathematical reasoning and code generation.
> > > > > > >
> > > > > > > We sincerely appreciate your suggestion—these additional experiments helped strengthen our empirical conclusions.
> > > > > > > We hope that these additional evaluations help further alleviate your concerns. We would be delighted to elaborate on any aspect of the method if you feel further clarification could be helpful.
> > > > > > >
> > > > > > >
> > > > > > > [1] Evaluating Large Language Models Trained on Code, OpenAI

---

### Official Review · Reviewer_QhYu · 2025-10-29

**Soundness:** 1
**Presentation:** 2
**Contribution:** 2
**Rating:** 4
**Confidence:** 4

**Summary:**

This paper discusses three key challenges in Generalized Reinforcement Policy Optimization (GRPO): gradient coupling between thoughts and answers, sparse reward signals due to limited parallel sampling, and unstable advantage estimation. To address these challenges, the authors introduce GRPO-MA, a straightforward yet theoretically sound method. GRPO-MA improves optimization efficiency and enhances the overall performance by generating multiple answers from each thought process.

**Strengths:**

1. This paper proposes the GRPO-MA algorithm, a simple but effective and general improvement
strategy for GRPO.
2. Multiple task domains are discussed, such as mathematics and code, with a focus on visual reasoning tasks.

**Weaknesses:**

1. Normalizing rewards can lead to zero advantages when rewards are all-zero, stalling optimization. The proposed method reduces multiple samplings of thoughts and increases the number of answers per thought. However, this could exacerbate the issue, as fewer thoughts with limited sampling may lead to invalid reasoning, especially when correct thoughts appear less frequently. The paper offers incremental innovation, limiting its overall contribution.

2. Contribution 1 mentions compatibility with other RL techniques such as DAPO, but the experiments do not validate its effectiveness in other methods, so this cannot be considered a contribution of the paper.
3. Similarly, the paper does not compare its results with DAPO, GRPO-CARE, or Dr.GRPO/GPG. Therefore, some of the issues mentioned in the challenges raise doubts about whether the proposed method truly addresses these challenges compared with other methods.
4. The experiments in this paper are conducted only on Qwen2.5-VL-3B-Ins, which has a small model size and lacks evidence for broader applicability. The improvements in results are minimal, with outcomes comparable to or even lower than those with TN=16. Additionally, the performance on code-related tasks is inferior to GRPO, which undermines the validity of Contribution 3.

**Questions:**

1. How does the reduction in multiple thought samplings and the increase in answer samples per thought impact optimization?
2. Is the proposed method effective when applied to other RL techniques like DAPO?
3. How does the proposed method compare to other approaches like DAPO, GRPO-CARE, or Dr.GRPO/GPG in addressing the identified challenges?
4. Is the performance of the proposed method consistent across other common model sizes  (such as 7B size)?

---

> ### Author Response · Authors · 2025-11-19
> **Response to Reviewer QhYu [1/5]**
>
> Thank you for your thoughtful review and for acknowledging the simplicity and cross-task effectiveness of our method. We appreciate your critical feedback, which has been invaluable in strengthening our paper.
>
> In response to your central concerns regarding advantage estimation, comparative baselines, and model scalability, we have provided additional theoretical analysis and conducted new experiments. We hope our responses and the new results will fully address your doubts and demonstrate the substantive contribution of our work.
>
> ## Major Clarification Before Point-by-Point Responses
>
> Before addressing the individual comments, we would like to respectfully clarify a major misunderstanding that appears to underlie several of your concerns (particularly in Weakness 1, Weakness 4, and Question 1).
>
> **Crucially, fair comparisons must be made under similar training-time and computational budgets.**
> This principle is central to understanding the intention behind our method. When training cost is held comparable, the appropriate baseline for evaluating GRPO-MA-T4A4 is GRPO-T4A1, not GRPO-T16A1 (TN=16).
>
> **However, several of your comments interpret GRPO-MA-T4A4 as a "reduced-thought version" of GRPO-T16A1 (TN=16) and compare these two settings as if they were designed to be direct counterparts.
> This interpretation does not reflect our method"s design and has unintentionally led to misjudgments regarding its contribution.**
>
> Our method does not aim to reduce the number of thoughts.
> The core idea of GRPO-MA is to sample multiple answers for each thought, improving advantage estimation while keeping the number of thoughts fixed. This is precisely why the method is named "GRPO with Multi-Answer" rather than "GRPO with Less-Thought".
>
> In this context, GRPO-T16A1 is included only as a super-strong baseline—a deliberately more expensive configuration—to further illustrate that GRPO-MA can even slightly surpass the stability benefits of extensive answer-level sampling while requiring significantly less training time.
>
> We hope this clarification provides the correct conceptual framing for the detailed point-by-point responses that follow.

---

> > ### Author Response · Authors · 2025-11-19
> > **Response to Reviewer QhYu [2/5]**
> >
> > ## W1.1
> > >Normalizing rewards can lead to zero advantages when rewards are all-zero, stalling optimization. The proposed method reduces multiple samplings of thoughts and increases the number of answers per thought. However, this could exacerbate the issue, as fewer thoughts with limited sampling may lead to invalid reasoning, especially when correct thoughts appear less frequently.
> >
> > Thank you very much for raising this important point. We sincerely apologize for any confusion caused by our earlier description of advantage value collapse.
> >
> > **Our intention is not to reduce the number of thoughts.**
> > Rather, **the central idea of our method is to sample multiple answers for each thought**.
> > We emphasize that this design choice is driven by the need to compare methods under similar training-time budgets. **Interpreting GRPO-MA-T4A4 as a "reduced-thought version" of GRPO-T16A1 would overlook the substantial difference in training time between the two settings.**
> >
> > In our experiments, GRPO-MA-T4A4 is primarily compared with GRPO-T4A1 **because the two settings require similar training time and computational resources**. We also include comparisons with GRPO-T16A1, which we treat as a **super-strong baseline** due to its substantially higher sampling cost. This super-strong baseline further helps us observe the performance of GRPO-MA and the benefits brought by multi-answer sampling.
> >
> > We fully agree that reducing the number of thoughts could lead to the issues, and we appreciate the opportunity to clarify that this is not the intent of our method.
> >
> > Importantly, the likelihood of advantage value collapse is strongly influenced by sampling time. More complete rollouts generally lower the risk of collapse, but also increase training cost. For this reason, directly comparing the collapse behavior of T4A4 and T16A1 may not be entirely fair, as T4A4 uses only about 60% of the sampling time of T16A1. Our goal is to strike a practical balance—reducing the probability of advantage collapse while keeping the additional sampling burden modest.
> >
> > Concretely, compared with GRPO-T4A1 (4 thoughts × 1 answer), our multi-answer variant GRPO-MA-T4A4 (4 thoughts × 4 answers) introduces only a small increase in sampling time (~10%), yet provides substantially richer reward signals. This leads to more stable advantage estimation and helps mitigate collapse (see Appendix A6.2, where T4A4 achieves the second-highest NonZeroRate; higher indicates less collapse). In this way, T4A4 attains stability close to that of T16A1 but with much lower computational cost, suggesting that multi-answer sampling is an efficient way to obtain diverse reward signals without incurring large overhead.
> >
> >
> >
> > ## W1.2
> > >The paper offers incremental innovation, limiting its overall contribution.
> >
> > Thank you very much for your thoughtful comment. We fully understand the concern that a method with a simple design might appear to offer limited innovation. However, we would like to clarify that simplicity does not mean a lack of innovation—in fact, when a simple mechanism helps alleviate several long-standing limitations, we believe such a design should be viewed as both practical and conceptually valuable.
> >
> > The novelty and contribution of our work center on **three key aspects**:
> >
> > **(1) A new unified perspective on GRPO.**
> > While prior studies have discussed several issues in GRPO individually, we are the first to consider **three fundamental challenges together**. This unified perspective reveals a deeper structural limitation of GRPO and **naturally motivates** answer-level multi-sampling as a coherent, conceptually grounded solution rather than an incremental modification.
> >
> > **(2) A new variance-based theoretical analysis.**
> > We introduce a **new theoretical tool** for understanding reasoning under GRPO: a variance analysis of advantage estimation based on multivariate delta method. This analysis uncovers a clear asymmetry between thought-level and answer-level sampling—
> >
> > * increasing **K** reduces variance only to a *non-zero floor*, whereas
> > * increasing **M** drives variance *provably toward zero*.
> >   This explains *why* answer-level multi-sampling stabilizes optimization and establishes it as a principled, theoretically justified design choice.
> >
> > **(3) Broad empirical validation across text, multimodal, and robotic tasks.**
> > We validate GRPO-MA across **pure-text reasoning**, **multimodal task**, and **simulator-based robotic manipulation**. This broad evaluation demonstrates that answer-level multi-sampling is widely applicable, extending far beyond the narrower settings explored in prior work.
> >
> > Taken together, these three components—**a unified conceptual perspective**, **a new variance-analysis framework**, and **extensive cross-domain validation**—form the core novelty and contribution of our work.
> > Overall, GRPO-MA is a simple yet non-trivial improvement—rooted in strong motivation, grounded in theory, and validated through broad empirical evidence.

---

> > > ### Author Response · Authors · 2025-11-19
> > > **Response to Reviewer QhYu [3/5]**
> > >
> > > ## W2
> > > >Contribution 1 mentions compatibility with other RL techniques such as DAPO, but the experiments do not validate its effectiveness in other methods, so this cannot be considered a contribution of the paper.
> > >
> > >
> > > Thank you very much for pointing out the ambiguity in our description of "compatibility" in Contribution 1 and for raising these important concerns. We appreciate the opportunity to clarify the intended meaning.
> > >
> > > DAPO and Dr.GRPO primarily focus on modifying the computation of advantage values or the loss function, whereas GRPO-MA operates at a different layer—its contribution lies in modifying the sampling method. Therefore, these approaches are complementary rather than mutually exclusive, and do not need to be treated as alternatives to each other. Indeed, several techniques from DAPO (e.g., clip-higher) are already incorporated into both our GRPO and GRPO-MA implementations to enhance training stability, **demonstrating in practice that DAPO and GRPO-MA are compatible**.
> > >
> > > We also realize that part of the confusion comes from an unintended ambiguity in our original description of Contribution 1. Your concern touches on two different concepts—effectiveness and compatibility—and we sincerely apologize for not making this distinction sufficiently clear. By "compatible," we intended to convey that GRPO-MA modifies GRPO at a different operational dimension and thus can be applied jointly with other GRPO enhancements, rather than claiming empirical validation of every possible combination.
> > >
> > > To avoid further misunderstanding, we will revise the description of Contribution 1 as follows: *We propose the GRPO-MA algorithm, a simple but effective and general improvement strategy for GRPO.*
> > >
> > > This revised phrasing avoids overclaiming while preserving the intended meaning. At the same time, the fact that we already employ DAPO techniques such as clip-higher in our implementation provides concrete evidence of the compatibility between GRPO-MA and existing GRPO variants.
> > >
> > > ## W3
> > > >Similarly, the paper does not compare its results with DAPO, GRPO-CARE, or Dr.GRPO/GPG. Therefore, some of the issues mentioned in the challenges raise doubts about whether the proposed method truly addresses these challenges compared with other methods.
> > >
> > >
> > > We sincerely appreciate your suggestion, as incorporating additional baselines can indeed enhance the completeness and persuasiveness of our experimental evaluation. Accordingly, we have added **GRPO-CARE**, **DAPO**, and **Dr.GRPO** as additional baselines.
> > > Below, we describe the specific implementation details for each method:
> > >
> > > * **GRPO-CARE**: strictly following the method described in the original paper, we added a *consistency reward* term.
> > > * **DAPO**: we removed the KL-divergence loss and incorporated *clip-higher*, *token-level policy gradient loss*, and *overlong reward shaping*. We did not include *Dynamic Sampling*, since in the trajectory prediction task the accuracy metric is a continuous value in ([0, 1]), rather than a discrete correctness signal.
> > > * **Dr.GRPO**: in accordance with the original paper, we modified the advantage computation and the optimization objective.
> > >
> > > Using these implementations, we report the updated results of all baselines on the trajectory prediction task:
> > >
> > > | Model     | DFD           | HD            | RMSE          | EndPoint      |
> > > |-----------|---------------|---------------|---------------|---------------|
> > > | GRPO      | 187.99        | 172.58        | 140.80        | 142.74        |
> > > | GRPO-CARE | 188.23        | 170.82        | 139.83        | 144.85        |
> > > | DAPO      | 184.08        | 167.75        | 136.18        | 146.99        |
> > > | Dr.GRPO   | 180.56        | 165.95        | 135.86        | 140.05        |
> > > | GRPO-MA | **151.10**   | **138.29**    | **111.59**    | **120.60**    |
> > >
> > > All experiments above are conducted with 4 thoughts per data sample, and GRPO-MA samples 4 answers for each thought. The results indicate that multi-answer sampling yields the strongest performance improvement.
> > > We hope that these additional results help address your concerns and provide a more comprehensive and balanced comparison.

---

> > > > ### Author Response · Authors · 2025-11-19
> > > > **Response to Reviewer QhYu [4/5]**
> > > >
> > > > ## W4.1
> > > > >The experiments in this paper are conducted only on Qwen2.5-VL-3B-Ins, which has a small model size and lacks evidence for broader applicability.
> > > >
> > > >
> > > > Thank you very much for highlighting this important point. We agree that evaluating larger model sizes is valuable for assessing the broader applicability of our method.
> > > >
> > > > Following your suggestion, we conducted additional experiments on the Qwen2.5-VL-7B-Instruct model. The updated results are shown below:
> > > >
> > > > | Model           | DFD        | HD         | RMSE       | EndPoint   |
> > > > | --------------- | ---------- | ---------- | ---------- | ---------- |
> > > > | GRPO-7B-T8A1    | 167.32     | 152.34     | 129.50     | 134.60     |
> > > > | GRPO-3B-MA-T4A4 | 151.10     | 138.29     | 111.59     | 120.60     |
> > > > | GRPO-7B-MA-T8A4 | **134.67** | **121.89** | **103.64** | **103.27** |
> > > >
> > > > As shown above, GRPO-MA remains consistently effective at the 7B scale and provides substantial improvements over the GRPO baseline. Due to the computational cost of running larger models—training the 7B model already requires 8×H100 GPUs with DeepSpeed ZeRO-3 for approximately three days—we are unfortunately unable to extend the experiments to even larger model sizes within the available time and resources. We sincerely apologize for this limitation. We also note that the 7B scale is widely adopted in recent GRPO-related studies (e.g., Dr.GRPO, GPG, and GRPO-CARE), and thus provides a practically relevant and commonly used setting for evaluating the broader applicability of GRPO-MA.
> > > >
> > > >
> > > > ## W4.2
> > > >
> > > > >The improvements in results are minimal, with outcomes comparable to or even lower than those with TN=16. Additionally, the performance on code-related tasks is inferior to GRPO, which undermines the validity of Contribution 3.
> > > >
> > > >
> > > > Thank you very much for raising this concern. We fully understand the importance of jointly evaluating absolute performance and computational efficiency when comparing GRPO-style methods.
> > > >
> > > > As we clarified in W1, the primary comparison for GRPO-MA-T4A4 is GRPO-T4A1, since they operate under similar training budgets. GRPO-T16A1 is included as a strong baseline rather than a direct competitor. Therefore, **directly comparing the raw performance gap between T4A4 and T16A1 would not be entirely fair**, as the two settings differ substantially in sampling time and computational cost.
> > > >
> > > > With this context, we analyze our results from both perspectives:
> > > >
> > > > 1. Absolute performance gain under equal thought-level sampling (K = 4).
> > > > When controlling for the same number of thoughts and under a comparable training-time budget, T4A4 provides a clear improvement over T4A1 by enriching the reward signal through multi-answer sampling and reducing the likelihood of advantage-value collapse.
> > > >
> > > > 2. Efficiency–performance trade-off relative to a strong baseline (T16A1). Although T16A1 samples more full responses, T4A4 achieves comparable or even slightly better results while requiring only about 60% of the sampling time. This demonstrates that multi-answer sampling is a more efficient way to obtain stable training signals.
> > > >
> > > > **Across 18 metrics** covering Math, Code, Object Detection, Affordance Prediction, Demand Prediction, OCR-based VQA, and Trajectory Prediction, **T4A4 outperforms T16A1 on 17 metrics**, with Code Pass@10 being the sole exception.
> > > >
> > > > In addition, we would like to emphasize that achieving even slightly better performance while using substantially less training time **offers meaningful practical value.** This result shows that the relationship between computational cost and final performance is not necessarily a strict trade-off—GRPO-MA achieves better outcomes under a smaller training budget. The perception that "the improvements are minimal" may stem from focusing solely on absolute performance, whereas a more comprehensive evaluation should also consider efficiency, where **improvements obtained with noticeably lower cost are particularly valuable in real-world large-scale training scenarios**.
> > > >
> > > > We sincerely apologize if our original wording of Contribution 3 did not sufficiently emphasize this efficiency–performance balance, which may have contributed to the perception that the improvements are minimal. We will revise the text to more clearly articulate our finding: T4A4 consistently improves upon T4A1 and achieves performance on par with or slightly better than T16A1, despite using substantially fewer computational resources.
> > > >
> > > > We hope this clarification addresses your concern and more accurately conveys the practical significance of our contributions.

---

> > > > > ### Author Response · Authors · 2025-11-19
> > > > > **Response to Reviewer QhYu [5/5]**
> > > > >
> > > > > ## Q1
> > > > > >How does the reduction in multiple thought samplings and the increase in answer samples per thought impact optimization?
> > > > >
> > > > >
> > > > > Thank you very much for the question. **Our intention is not to reduce the number of thoughts.**
> > > > > Rather, **the central idea of our method is to sample multiple answers for each thought**.
> > > > > We emphasize that this design choice is driven by the need to compare methods under similar training-time budgets. **Interpreting GRPO-MA-T4A4 as a "reduced-thought version" of GRPO-T16A1 would overlook the substantial difference in training time between the two settings.**
> > > > >
> > > > > GRPO-MA does not modify the underlying optimization rule of GRPO. Instead, it replaces the original advantage estimator with a lower-variance and more reliable multi-answer estimate, which is then used directly within the standard GRPO objective. This refined estimator influences optimization in the following ways:
> > > > >
> > > > > 1. **Mitigated gradient coupling.**
> > > > >    Multiple answers generated from the same thought are averaged to produce a more stable reward signal. This reduces noise caused by mismatches between individual thoughts and answers, leading to more reliable thought-level gradients.
> > > > >
> > > > > 2. **Lower-variance advantage estimation.**
> > > > >    Evaluating each thought with multiple answers provides a more accurate estimate of its value, resulting in reduced variance and improved optimization stability.
> > > > >
> > > > > 3. **Better sampling efficiency.**
> > > > >    GRPO-MA enriches the reward signal without proportional increases in sampling time. Compared to sampling (K \times M) full responses, generating (K) thoughts and sampling multiple answers per thought adds only modest overhead while substantially reducing the risk of advantage-value collapse. This design achieves nearly the same stability benefits as using many more full rollouts, but with far lower computational cost.
> > > > >
> > > > >
> > > > >
> > > > > ## Q2+Q3
> > > > > >Is the proposed method effective when applied to other RL techniques like DAPO?
> > > > > >How does the proposed method compare to other approaches like DAPO, GRPO-CARE, or Dr.GRPO/GPG in addressing the identified challenges?
> > > > >
> > > > > Thank you very much for your question!
> > > > > Please refer to the more detailed baseline comparison experiments provided in W2+W3.
> > > > >
> > > > > ## Q4
> > > > > >Is the performance of the proposed method consistent across other common model sizes (such as 7B size)?
> > > > >
> > > > > Thank you very much for your question!
> > > > >
> > > > > Please refer to the more detailed scaling experiments provided in W4.1

---

### Official Review · Reviewer_JCTn · 2025-10-31

**Soundness:** 2
**Presentation:** 3
**Contribution:** 2
**Rating:** 4
**Confidence:** 4

**Summary:**

This paper proposes GRPO-MA, an extension of Group Relative Policy Optimization for Chain-of-Thought training. The method generates multiple answers per thought to reduce gradient variance, decouple thought–answer updates, and stabilize training. Theoretically, it shows that more answers yield lower variance; empirically, GRPO-MA improves stability and performance across math, code, and multimodal tasks.

**Strengths:**

1. **Extensive experiments**: The paper evaluates GRPO-MA on a wide range of reasoning and multimodal tasks (math, code, vision, and manipulation), demonstrating consistent gains in both performance and training stability.
2. **Strong theoretical grounding**: The derivation using the delta method clearly connects multi-answer sampling with variance reduction in advantage estimation, providing a solid mathematical basis for the approach.
3. **Cross-modal applicability**: The method generalizes well across text and vision tasks, highlighting its robustness and potential for broad adoption in reinforcement learning for reasoning.

**Weaknesses:**

1. **Overly long introduction**: The introduction is too long and repeats background information that could be moved to the related work section for better focus and readability.
2. **Limited novelty**: The proposed multi-answer strategy is intuitive and builds naturally on GRPO. While it’s a solid and useful extension, it feels more like a practical enhancement than a major conceptual change.
3. **Limited model diversity**: The experiments focus mainly on the Qwen2.5-VL family, leaving it unclear whether the proposed GRPO-MA method generalizes equally well to other model architectures or sizes.
4. **Lack of scaling analysis**: The paper would benefit from an experiment exploring how GRPO-MA scales with model size and training time compared to the original GRPO. Such a scaling law study could clarify the method’s efficiency and practicality for larger models.
5. **Missing comparisons with other RL baselines**: While the paper mentions several GRPO variants (e.g., DAPO, Dr.GRPO, GPG), it only compares GRPO-MA against the original GRPO. Including results or discussion on how GRPO-MA performs relative to these methods would provide a clearer picture of its advantages and limitations.

**Questions:**

Please refer to weaknesses

---

> ### Author Response · Authors · 2025-11-19
> **Response to Reviewer JCTn [1/3]**
>
> We are deeply grateful for your thoughtful and constructive review of our manuscript. We sincerely appreciate you recognizing the extensive experimental validation across diverse tasks, the strong theoretical grounding provided by the delta method derivation, and the demonstrated cross-modal applicability of GRPO-MA. Your insightful summary of our core contributions—reducing gradient variance and decoupling thought-answer updates—is particularly encouraging.
>
> In response to your valuable feedback, we have prepared the following point-by-point responses and have conducted additional experiments where suggested. We hope that our explanations and new results will fully address your concerns and further substantiate the value and robustness of our proposed method.
>
> ## W1
> >Overly long introduction: The introduction is too long and repeats background information that could be moved to the related work section for better focus and readability.
>
> Thank you very much for this constructive suggestion. We appreciate your careful reading of our work and apologize for any redundancy in the current introduction. Following your suggestion, we have revised the Introduction by removing repeated background material and streamlining the presentation while preserving the essential motivation and context. We hope that these revisions help make the section clearer and easier to follow.
>
>
> ## W2
> >Limited novelty: The proposed multi-answer strategy is intuitive and builds naturally on GRPO. While it's a solid and useful extension, it feels more like a practical enhancement than a major conceptual change.
>
> Thank you very much for your thoughtful comment and for recognizing our method as a solid and useful extension of GRPO. We sincerely appreciate this positive assessment. We fully understand the concern that a method with a simple design might appear to offer limited innovation. However, we would like to clarify that simplicity does not mean a lack of innovation—in fact, when a simple mechanism helps alleviate several long-standing limitations, we believe such a design should be viewed as both practical and conceptually valuable.
>
> The novelty and contribution of our work center on **three key aspects**:
>
> **(1) A new unified perspective on GRPO.**
> While prior studies have discussed several issues in GRPO individually, we are the first to consider **three fundamental challenges together**. This unified perspective reveals a deeper structural limitation of GRPO and **naturally motivates** answer-level multi-sampling as a coherent, conceptually grounded solution rather than an incremental modification.
>
> **(2) A new variance-based theoretical analysis.**
> We introduce a **new theoretical tool** for understanding reasoning under GRPO: a variance analysis of advantage estimation based on multivariate delta method. This analysis uncovers a clear asymmetry between thought-level and answer-level sampling—
>
> * increasing **K** reduces variance only to a *non-zero floor*, whereas
> * increasing **M** drives variance *provably toward zero*.
>   This explains *why* answer-level multi-sampling stabilizes optimization and establishes it as a principled, theoretically justified design choice.
>
> **(3) Broad empirical validation across text, multimodal, and robotic tasks.**
> We validate GRPO-MA across **pure-text reasoning**, **multimodal task**, and **simulator-based robotic manipulation**. This broad evaluation demonstrates that answer-level multi-sampling is widely applicable, extending far beyond the narrower settings explored in prior work.
>
> Taken together, these three components—**a unified conceptual perspective**, **a new variance-analysis framework**, and **extensive cross-domain validation**—form the core novelty and contribution of our work.
> Overall, GRPO-MA is a simple yet non-trivial improvement—rooted in strong motivation, grounded in theory, and validated through broad empirical evidence.

---

> > ### Author Response · Authors · 2025-11-19
> > **Response to Reviewer JCTn [2/3]**
> >
> > ## W3
> >
> > >Limited model diversity: The experiments focus mainly on the Qwen2.5-VL family, leaving it unclear whether the proposed GRPO-MA method generalizes equally well to other model architectures or sizes.
> >
> >
> > Thank you for your valuable comment.
> >
> > Qwen models are currently among the most widely adopted open-source foundation models, covering a broad range of modalities and model sizes. Many top-ranking open-source LLMs on the Hugging Face Open LLM Leaderboard are either derived from or fine-tuned on Qwen, indicating its central role in the open-source community. Therefore, using Qwen-based models follows prevailing practice in both academic and open-source settings.
> >
> > However, we fully agree that evaluating only on a Qwen's VLM architecture is insufficient to demonstrate the generalizability of GRPO-MA. To address this concern, we further include experiments on a pure text-only LLM—Qwen3-4B-Instruct—under the same Math and Code settings used in the main paper. The results are summarized below:
> >
> > | Model        | Pass@1 on Math | Pass@1 on Code |
> > | ------------ | -------------- | -------------- |
> > | GRPO-T4A1    | 17.53          | 4.22           |
> > | GRPO-MA-T4A4 | **18.07**      | **6.25**       |
> >
> > These results show consistent improvements across both modalities (VLM vs LLM) and different model architectures, suggesting that GRPO-MA is not tied to a specific model type.
> >
> > As elaborated in W4, we additionally evaluate GRPO-MA on a larger-scale VLM, Qwen2.5-VL-7B-Instruct, and again observe stable gains. Together with the above results, this provides empirical evidence that GRPO-MA generalizes across architectures (LLM/VLM) and across model sizes (3B → 7B).
> >
> > We hope these additional experiments alleviate concerns regarding model diversity and further validate the broad applicability of GRPO-MA.
> >
> >
> > ## W4
> > >Lack of scaling analysis: The paper would benefit from an experiment exploring how GRPO-MA scales with model size and training time compared to the original GRPO. Such a scaling law study could clarify the method's efficiency and practicality for larger models.
> >
> > We greatly appreciate your suggestion regarding larger model sizes.
> > We have conducted trajectory prediction experiments on the Qwen2.5-VL-7B-Instruct model. The results are shown below:
> >
> > | Model           | DFD        | HD         | RMSE       | EndPoint   |
> > | --------------- | ---------- | ---------- | ---------- | ---------- |
> > | GRPO-7B-T8A1    | 167.32     | 152.34     | 129.50     | 134.60     |
> > | GRPO-3B-MA-T4A4 | 151.10     | 138.29     | 111.59     | 120.60     |
> > | GRPO-7B-MA-T8A4 | **134.67** | **121.89** | **103.64** | **103.27** |
> >
> > We find that our method remains effective on the 7B model. Unfortunately, we are unable to extend experiments to even larger models. Running the 7B model already requires 8×H100 GPUs with DeepSpeed ZeRO-3 for approximately three days, and scaling further is beyond our available time and computational budget. We sincerely apologize for these limitations, and we hope the above 7B results provide a meaningful reference for assessing the broader applicability of our method. We further note that the 7B model is a widely adopted scale in recent academic studies (e.g., Dr.GRPO, GPG, and GRPO-CARE), and we hope these results serve as a helpful reference for evaluating the broader applicability of our method.
> >
> > For the scaling analysis with respect to training time, this experiment is still running due to the substantial computational cost involved. We will supplement this experiment as soon as possible.

---

> > > ### Author Response · Authors · 2025-11-19
> > > **Response to Reviewer JCTn [3/3]**
> > >
> > > ## W5
> > > >Missing comparisons with other RL baselines: While the paper mentions several GRPO variants (e.g., DAPO, Dr.GRPO, GPG), it only compares GRPO-MA against the original GRPO. Including results or discussion on how GRPO-MA performs relative to these methods would provide a clearer picture of its advantages and limitations.
> > >
> > >
> > > We sincerely appreciate your suggestion, as incorporating additional baselines can indeed enhance the completeness and persuasiveness of our experimental evaluation. Accordingly, we have added **GRPO-CARE**, **DAPO**, and **Dr.GRPO** as additional baselines.
> > > Below, we describe the specific implementation details for each method:
> > >
> > > * **GRPO-CARE**: strictly following the method described in the original paper, we added a *consistency reward* term.
> > > * **DAPO**: we removed the KL-divergence loss and incorporated *clip-higher*, *token-level policy gradient loss*, and *overlong reward shaping*. We did not include *Dynamic Sampling*, since in the trajectory prediction task the accuracy metric is a continuous value in ([0, 1]), rather than a discrete correctness signal.
> > > * **Dr.GRPO**: in accordance with the original paper, we modified the advantage computation and the optimization objective.
> > >
> > > Using these implementations, we report the updated results of all baselines on the trajectory prediction task:
> > >
> > > | Model     | DFD           | HD            | RMSE          | EndPoint      |
> > > |-----------|---------------|---------------|---------------|---------------|
> > > | GRPO      | 187.99        | 172.58        | 140.80        | 142.74        |
> > > | GRPO-CARE | 188.23        | 170.82        | 139.83        | 144.85        |
> > > | DAPO      | 184.08        | 167.75        | 136.18        | 146.99        |
> > > | Dr.GRPO   | 180.56        | 165.95        | 135.86        | 140.05        |
> > > | GRPO-MA | **151.10**   | **138.29**    | **111.59**    | **120.60**    |
> > >
> > > All experiments above are conducted with 4 thoughts per data sample, and GRPO-MA samples 4 answers for each thought. The experimental results show that multi-answer sampling provides the greatest improvement for GRPO on the trajectory prediction task.
> > > We hope that these updated results address your concerns and provide a more comprehensive comparison.

---

> > > > ### Author Response · Authors · 2025-11-23
> > > > **Training-Time Scaling Analysis**
> > > >
> > > > Thank you very much for the suggestion to include a scaling analysis of the training time.
> > > >
> > > > We have expanded our trajectory-prediction experiments to compare GRPO-MA and GRPO under increased training volume. Specifically, we doubled the training duration for both methods—from 1 epoch (3000 steps) to 2 epochs (6000 steps). Here, the suffix "3000" and "6000" denote the total number of training steps. The results are summarized below:
> > > >
> > > > | Model        | DFD        | HD         | RMSE       | EndPoint   |
> > > > |--------------|------------|------------|------------|------------|
> > > > | GRPO-3000    | 187.99     | 172.58     | 140.80     | 142.74     |
> > > > | GRPO-6000    | 167.52     | 157.21     | 109.03     | 122.56     |
> > > > | GRPO-MA-3000 | 151.10     | 138.29     | 111.59     | 120.60     |
> > > > | GRPO-MA-6000 | **138.44** | **125.77** | **104.59** | **105.31** |
> > > >
> > > > All experiments above use 4 thoughts per data sample, and GRPO-MA samples 4 answers for each thought. As shown above, both GRPO-MA and GRPO improve when trained longer, and GRPO-MA continues to show stronger results under the same extended training budget.

---

### Official Review · Reviewer_Q536 · 2025-11-03

**Soundness:** 3
**Presentation:** 3
**Contribution:** 2
**Rating:** 4
**Confidence:** 4

**Summary:**

The paper proposes GRPO-MA, a variant of GRPO that samples M answers per thought, uses the average reward per thought to compute a thought-level advantage, and applies token-level PPO-style updates to both thoughts and answers. Theoretically, the authors argue (via the delta method) that increasing M monotonically reduces the variance of the thought advantage; empirically, they show higher accuracy, fewer gradient spikes, and better sample-efficiency across math, code, vision, and an embodied manipulation task.

**Strengths:**

1. The paper is generally well-written and structured, followed by the motivation and explanation.
2. Multi-answer sampling decouples noisy answer outcomes from thought quality and directly generates reward signals.
3. Experiments span math/code and diverse multimodal tasks with consistent gains.

**Weaknesses:**

1. The only difference I can think of between multi-answer with one rollout and single-answer generation is that, in the former, at least one answer among the multiple generations is correct. However, how can the authors guarantee that, given the same instruction and thinking trajectory, the model can generate the final answer directly without rethinking? It would be helpful to include an example illustrating this process.
2. This method can apply to both LLMs and VLMs. Why did the authors not evaluate it on text-only LLMs? One concern is that multimodal models produce shorter CoTs compared to LLMs, which makes the reasoning process simpler. It would be better to test the approach on longer reasoning tasks using LLMs.
3. It's better to add stronger baselines, such as DAPO, for a more comprehensive comparison.
4. Although the authors evaluate different values of $K$ and $M$, since these are the two most critical hyperparameters, I suggest conducting a more extensive exploration to determine the optimal combination of $K$ and $M$.

**Questions:**

1. The authors train a 3B model using LoRA. I suggest testing with a larger base model with LoRA.
2. In Figure 3, the performance appears to decrease slightly when $M = 6$ and then increase afterward. Could the authors explain the reason behind this?

---

> ### Author Response · Authors · 2025-11-19
> **Response to Reviewer Q536 [1/3]**
>
> We sincerely thank you for your thorough and insightful review of our paper. We are particularly grateful for your positive assessment of our work's strengths, especially your recognition of its soundness, clear presentation, and the effectiveness of our core contribution—using multi-answer sampling to decouple noisy outcomes from thought quality. Your acknowledgment that the method "directly generates reward signals" and that our experiments "span diverse tasks with consistent gains" is greatly encouraging.
>
> In response to the questions you raised, we have prepared the following clarifications and additional experiments. We hope our explanations and new results will adequately address your concerns and further demonstrate the robustness and generality of our proposed GRPO-MA method.
>
> ## W1
> >The only difference I can think of between multi-answer with one rollout and single-answer generation is that, in the former, at least one answer among the multiple generations is correct. However, how can the authors guarantee that, given the same instruction and thinking trajectory, the model can generate the final answer directly without rethinking? It would be helpful to include an example illustrating this process.
>
>
> Thank you very much for pointing out the need for clarification regarding our multi-answer generation process.
>
> To illustrate this process, we provide an example using a multimodal task:
>
> - Step 1: Given a prompt—comprising both textual instructions and an image—we first generate a single chain-of-thought response with a Vision-Language Model (VLM), using `</think>` as the `stop_strings`  (without including the final answer). When the model produces the `</think>` token, generation automatically stops.
>
> - Step 2: We then concatenate the initial prompt with the generated chain-of-thought, duplicate this combined input (M) times, and feed them back into the VLM to generate answers under a high-temperature sampling setting. In our experiments, setting the temperature to 1 leads to diverse outputs, with each generation often producing a different answer.
>
> **We provide the `grpo_ma_trainer.py` code in the supplementary materials. You may refer to the `_generate_and_score_completions_need_gather` function for more detailed information on the multi-answer generation process.**
>
>
> *PS: `stop_strings` is a parameter in `GenerationConfig` of HuggingFace transformers library.*
>
> This high-temperature multi-sampling introduces variability in the M answers, enriching the reward signal. **As shown in Appendix A6.2**, compared to GRPO-T4A1, GRPO-MA-T4A4 significantly increases the probability that "at least one answer among the multiple generations is correct" (referred to as NonZeroRate in our paper). While we cannot guarantee that the correct answer will always be sampled, GRPO-MA substantially increases this probability compared to GRPO-T4A1. Although GRPO-MA-T4A4 does not surpass GRPO-T16A1 in NonZeroRate, it achieves a comparable level while using only about 60% of the sampling time. Therefore, the key advantage of GRPO-MA lies in achieving higher NonZeroRate with less computation, leading to richer reward signals.
>
> We note a potential concern: if the model fails to produce the `</think>` token in Step 1, generation would continue past the intended stopping point, preventing Step 2 from proceeding as designed. In practice, such responses are considered to violate the required format and are penalized by the format reward. During training across multiple tasks, we observed that the formatting reward successfully guided the model to learn the correct formatting within approximately 100 steps, enabling smooth multi-answer sampling.

---

> > ### Author Response · Authors · 2025-11-19
> > **Response to Reviewer Q536 [2/3]**
> >
> > ## W2
> > >This method can apply to both LLMs and VLMs. Why did the authors not evaluate it on text-only LLMs? One concern is that multimodal models produce shorter CoTs compared to LLMs, which makes the reasoning process simpler. It would be better to test the approach on longer reasoning tasks using LLMs.
> >
> >
> > Thank you very much for your thoughtful suggestion and for highlighting the potential interest in evaluating our method on text-only LLMs.
> > To maintain consistency across multiple tasks, we primarily used the same model in our experiments. However, we observed that the length of the thought chains generated by the VLM is more dependent on the task itself than on the modality: code and math tasks tend to produce significantly longer chains of thought compared to multimodal tasks.
> >
> > To further address your concern, we conducted additional experiments using the Qwen3-4B-Instruct model on Math and Code tasks:
> >
> > | Model        | Pass@1 on Math | Pass@1 on Code |
> > |--------------|----------------|----------------|
> > | GRPO-T4A1    | 17.53          | 4.22          |
> > | GRPO-MA-T4A4 | **18.07**      | **6.25**      |
> >
> > The experimental setup remains consistent with the paper. Since the Qwen3-4B-Instruct model is relatively strong, we replace the original metrics with **Pass@1**.
> >
> > These results indicate that GRPO-MA continues to provide improvements even on pure text models, supporting the generality of our approach.
> >
> >
> >
> > ## W3
> > >It's better to add stronger baselines, such as DAPO, for a more comprehensive comparison.
> >
> >
> > Thank you very much for your valuable suggestion regarding adding stronger baselines.
> >
> > DAPO, Dr.GRPO focus on modifying the calculation of advantage values and loss functions, whereas our algorithm primarily modifies the sampling process. Therefore, we view them as complementary rather than mutually exclusive.
> >
> > We sincerely appreciate your suggestion, as incorporating additional baselines can indeed enhance the completeness and persuasiveness of our experimental evaluation. Accordingly, we have added **GRPO-CARE**, **DAPO**, and **Dr.GRPO** as additional baselines.
> > Below, we describe the specific implementation details for each method:
> >
> > * **GRPO-CARE**: strictly following the method described in the original paper, we added a *consistency reward* term.
> > * **DAPO**: we removed the KL-divergence loss and incorporated *clip-higher*, *token-level policy gradient loss*, and *overlong reward shaping*. We did not include *Dynamic Sampling*, since in the trajectory prediction task the accuracy metric is a continuous value in ([0, 1]), rather than a discrete correctness signal.
> > * **Dr.GRPO**: in accordance with the original paper, we modified the advantage computation and the optimization objective.
> >
> > Using these implementations, we report the updated results of all baselines on the trajectory prediction task:
> >
> > | Model     | DFD           | HD            | RMSE          | EndPoint      |
> > |-----------|---------------|---------------|---------------|---------------|
> > | GRPO      | 187.99        | 172.58        | 140.80        | 142.74        |
> > | GRPO-CARE | 188.23        | 170.82        | 139.83        | 144.85        |
> > | DAPO      | 184.08        | 167.75        | 136.18        | 146.99        |
> > | Dr.GRPO   | 180.56        | 165.95        | 135.86        | 140.05        |
> > | GRPO-MA   | **151.10**   | **138.29**    | **111.59**    | **120.60**    |
> >
> > All experiments above are conducted with 4 thoughts per data sample, and GRPO-MA samples 4 answers for each thought. The experimental results show that multi-answer sampling provides the greatest improvement for GRPO on the trajectory prediction task.
> > We hope that these updated results address your concerns and provide a more comprehensive comparison.

---

> > > ### Author Response · Authors · 2025-11-19
> > > **Response to Reviewer Q536 [3/3]**
> > >
> > > ## W4
> > > >Although the authors evaluate different values of K and M, since these are the two most critical hyperparameters, I suggest conducting a more extensive exploration to determine the optimal combination of K and M.
> > >
> > >
> > > Thank you very much for your valuable suggestion regarding a more extensive exploration of the hyperparameters K and M.
> > >
> > > From our theoretical derivation, K and M affect the variance **independently**. Specifically, increasing M directly reduces the variance of each thought's advantage estimate (since $\mathrm{Var[A(th_i)]}\propto 1/M$), while increasing K introduces more diverse thoughts. This independence allows us to analyze the two hyperparameters separately when determining their appropriate ranges.
> > >
> > > Consistent with this analysis, our theoretical results show that **the rate of variance reduction slows as M increases**, and our ablations also confirm that when M > 4, the performance gain becomes marginal. In practice, this means that increasing M beyond 4 provides little additional benefit while incurring extra computational cost, which is why we adopt M = 4 as a reasonable and efficient choice across all tasks.
> > >
> > > Regarding K, its optimal value largely depends on available computational resources. In principle, K can be as large as computationally feasible, since a larger K provides richer and more diverse thought candidates.
> > >
> > > We believe that generating **more diverse thoughts** and **accurately estimating each thought's advantage** both play important roles in effective model optimization.
> > >
> > >
> > > ## Q1:
> > > >The authors train a 3B model using LoRA. I suggest testing with a larger base model with LoRA.
> > >
> > >
> > > We greatly appreciate your suggestion regarding larger model sizes.
> > > We have conducted trajectory prediction experiments on the Qwen2.5-VL-7B-Instruct model. The results are shown below:
> > >
> > > | Model           | DFD        | HD         | RMSE       | EndPoint   |
> > > | --------------- | ---------- | ---------- | ---------- | ---------- |
> > > | GRPO-7B-T8A1    | 167.32     | 152.34     | 129.50     | 134.60     |
> > > | GRPO-3B-MA-T4A4 | 151.10     | 138.29     | 111.59     | 120.60     |
> > > | GRPO-7B-MA-T8A4 | **134.67** | **121.89** | **103.64** | **103.27** |
> > >
> > > We find that our method remains effective on the 7B model. Unfortunately, we are unable to extend experiments to even larger models. Running the 7B model already requires 8×H100 GPUs with DeepSpeed ZeRO-3 for approximately three days, and scaling further is beyond our available time and computational budget. We sincerely apologize for these limitations, and we hope the above 7B results provide a meaningful reference for assessing the broader applicability of our method. We further note that the 7B model is a widely adopted scale in recent academic studies (e.g., Dr.GRPO, GPG, and GRPO-CARE), and we hope these results serve as a helpful reference for evaluating the broader applicability of our method.
> > >
> > >
> > >
> > >
> > > ## Q2
> > > >In Figure 3, the performance appears to decrease slightly when M=6 and then increase afterward. Could the authors explain the reason behind this?
> > >
> > >
> > > We think the slight performance drop observed in T4A6 may be due to stochastic factors during training. In our ablation study, we used only a single random seed to balance computational resources across the main experiments on multiple diverse tasks, which can occasionally lead to minor fluctuations. We apologize if our original explanation was not sufficiently clear, and we hope this clarification helps to better contextualize the observed behavior.

---

### Author Response · Authors · 2025-11-19
**Common Response [1/3]**

We sincerely thank all reviewers for the time, effort, and care you dedicated to evaluating our submission. The insightful comments, constructive suggestions, and thoughtful questions you provided have substantially strengthened the clarity and rigor of our work. We would also like to express our deep appreciation to the Area Chair (AC) for the thoughtful oversight and careful coordination you have provided throughout the process.

We are especially grateful because we understand that the reviewers and AC handling this submission, may also navigate a rebuttal period—balancing the preparation of your own papers while still investing considerable time to review and offer meaningful feedback on ours. We deeply appreciate the commitment and attention you devoted to our work during such a busy and high-pressure time.

We genuinely value the opportunity to learn from your feedback and to engage in this exchange with the reviewers and AC. Your guidance has been truly invaluable, and we will continue to refine our work with great care.

We greatly appreciate the reviewers' recognition of our paper's writing quality, motivation, theoretical analysis, and the breadth of our experimental evaluation. "The paper is generally well-written and structured, followed by the motivation and explanation."(Q536), "The paper evaluates GRPO-MA on a wide range of reasoning and multimodal tasks..., demonstrating consistent gains in both performance and training stability."(JCTn), "...using the delta method clearly connects multi-answer sampling with variance reduction in advantage estimation.."(JCTn), "Multiple task domains are discussed, such as mathematics and code, with a focus on visual reasoning tasks." (QhYu) "The paper articulates three concrete challenges...are well-motivated in both text and multimodal tasks."(5d7r) "The variance analysis using the delta method is both non-trivial and practically relevant."(5d7r)

We have also noted the reviewers' valuable suggestions and potential concerns regarding our submission. Below, we provide responses to several common questions that appeared across multiple reviews.

### **Note: We have included our complete training code and the trajectory prediction dataset in the supplementary material to facilitate reproducibility verification.**

## Common Question 1 (CQ1)
> Lack of stronger baselines such as DAPO, GRPO-CARE, and Dr.GRPO

We sincerely appreciate the reviewers' suggestion, as incorporating additional baselines can indeed enhance the completeness and persuasiveness of our experimental evaluation. Accordingly, we have added **GRPO-CARE**, **DAPO**, and **Dr.GRPO** as additional baselines.
Below, we describe the specific implementation details for each method:

* **GRPO-CARE**: strictly following the method described in the original paper, we added a *consistency reward* term.
* **DAPO**: we removed the KL-divergence loss and incorporated *clip-higher*, *token-level policy gradient loss*, and *overlong reward shaping*. We did not include *Dynamic Sampling*, since in the trajectory prediction task the accuracy metric is a continuous value in ([0, 1]), rather than a discrete correctness signal.
* **Dr.GRPO**: in accordance with the original paper, we modified the advantage computation and the optimization objective.

Using these implementations, we report the updated results of all baselines on the trajectory prediction task:

| Model     | DFD        | HD         | RMSE       | EndPoint   |
| --------- | ---------- | ---------- | ---------- | ---------- |
| GRPO      | 187.99     | 172.58     | 140.80     | 142.74     |
| GRPO-CARE | 188.23     | 170.82     | 139.83     | 144.85     |
| DAPO      | 184.08     | 167.75     | 136.18     | 146.99     |
| Dr.GRPO   | 180.56     | 165.95     | 135.86     | 140.05     |
| GRPO-MA   | **151.10** | **138.29** | **111.59** | **120.60** |

We apologize that due to limitations in time and computational resources, we were only able to add these baseline comparisons for the trajectory prediction task. We hope that these additional results help alleviate your concerns and further demonstrate the robustness of our empirical evaluation.

---

> ### Author Response · Authors · 2025-11-19
> **Common Response [2/3]**
>
> ## Common Question 2 (CQ2)
> > Lack of method validation on larger models and different model architectures.
>
> We greatly appreciate the reviewers' suggestion regarding larger model sizes and different model architectures.
> To address this, we conduct trajectory-prediction experiments on the larger-scale vision–language model Qwen2.5-VL-7B-Instruct, and we additionally evaluate GRPO-MA on the pure-text language model with a different architecture, Qwen3-4B-Instruct, for math and code reasoning tasks.
>
> The results are shown below:
> #### Trajectory Prediction
> | Model           | DFD        | HD         | RMSE       | EndPoint   |
> | --------------- | ---------- | ---------- | ---------- | ---------- |
> | GRPO-7B-T8A1    | 167.32     | 152.34     | 129.50     | 134.60     |
> | GRPO-3B-MA-T4A4 | 151.10     | 138.29     | 111.59     | 120.60     |
> | GRPO-7B-MA-T8A4 | **134.67** | **121.89** | **103.64** | **103.27** |
>
> We find that our method remains effective on the 7B model. Due to time and computational limitations, we regret that we are unable to extend experiments to even larger models at this stage. Nevertheless, the 7B model is a standard size widely used in academic research (e.g., in studies such as Dr.GRPO, GPG, and GRPO-CARE), and we hope that these results provide an informative reference regarding the potential effectiveness of our approach.
>
> #### Math&Code
>
> | Model        | Pass@1 on Math | Pass@1 on Code |
> |--------------|----------------|----------------|
> | GRPO-T4A1    | 17.53          | 4.22          |
> | GRPO-MA-T4A4 | **18.07**      | **6.25**      |
>
> These results indicate that GRPO-MA continues to provide improvements even on pure text models,  Qwen3-4B-Instruct, supporting the generality of our approach on different model architectures.
>
>
> ## Common Question 3 (CQ3)
>
> > Lack of novelty and limited contribution
>
> We thank the reviewers for their thoughtful comments regarding the novelty of our method. We fully understand the concern that a method with a simple design might appear to offer limited innovation. However, we would like to clarify that simplicity does not mean a lack of innovation—in fact, when a simple mechanism helps alleviate several long-standing limitations, we believe such a design should be viewed as both practical and conceptually valuable.
>
> The novelty and contribution of our work center on **three key aspects**:
>
> **(1) A new unified perspective on GRPO.**
> While prior studies have discussed several issues in GRPO individually, we are the first to consider **three fundamental challenges together**. This unified perspective reveals a deeper structural limitation of GRPO and naturally motivates answer-level multi-sampling as a coherent, conceptually grounded solution rather than an incremental modification.
>
> **(2) A new variance-based theoretical analysis.**
> We introduce a **new theoretical tool** for understanding reasoning under GRPO: a variance analysis of advantage estimation based on multivariate delta method. This analysis uncovers a clear asymmetry between thought-level and answer-level sampling—
>
> * increasing **K** reduces variance only to a *non-zero floor*, whereas
> * increasing **M** drives variance *provably toward zero*.
>   This explains *why* answer-level multi-sampling stabilizes optimization and establishes it as a principled, theoretically justified design choice.
>
> **(3) Broad empirical validation across text, multimodal, and robotic tasks.**
> We validate GRPO-MA across **pure-text reasoning**, **multimodal task**, and **simulator-based robotic manipulation**. This broad evaluation demonstrates that answer-level multi-sampling is widely applicable, extending far beyond the narrower settings explored in prior work.
>
> Taken together, these three components—**a unified conceptual perspective**, **a new variance-analysis framework**, and **extensive cross-domain validation**—form the core novelty and contribution of our work.
> Overall, GRPO-MA is a simple yet non-trivial improvement—rooted in strong motivation, grounded in theory, and validated through broad empirical evidence.

---

> > ### Author Response · Authors · 2025-11-19
> > **Common Response [3/3]**
> >
> > ## Common Question 4 (CQ4)
> > > How to make sure that the same thought can yield multiple different answers?
> >
> > We thank the reviewers for raising their concerns regarding sampling methods.
> >
> > In GRPO-MA, even when the thought is fixed, using high-temperature sampling during answer generation **naturally** produces diverse final answer with high probability. This diversity is crucial for enriching reward signals.
> >
> > We apologize that the original draft did not make the empirical support sufficiently explicit.
> > **In fact, we have already conducted statistical analyses to verify that a single thought can lead to multiple distinct answers.** As discussed in **Appendix A.6.2**, we report the NonZeroRate—the proportion of samples for which at least one of the generated answers is correct.
> > If a single thought were not capable of producing diverse answers, then the NonZeroRate values of **GRPO-T4A1** and **GRPO-MA-T4A4** would be nearly identical. However, our experimental results show that GRPO-MA-T4A4 achieves a noticeably higher NonZeroRate than GRPO-T4A1, suggesting that one thought indeed produces multiple distinct outputs in practice.
> >
> > We hope this clarifies why a single thought can yield multiple answers under GRPO-MA.
> >
> >
> > ## Paper Revision and Plan
> >
> > Below we summarize the nine revisions we have already made to our paper:
> >
> > 1. Following reviewer JCTn's suggestion, we shorten the Introduction by removing redundant background material while preserving all essential motivation and context.
> >
> > 2. Following reviewer 5d7r's recommendation, we rewrite the GRPO and GRPO-MA formulations. The original Equations (1), (2), and (3) are now consolidated and rewritten as the new Equations (1) and (2). We also reformat the task descriptions in Section 5.1.1 into a clean, well-structured table.
> >
> > 3. As suggested by reviewer 5d7r, we reduce the length of the GRPO algorithm description to improve clarity and focus. Specifically, we revise the description of the second challenge to emphasize the trade-off between reward richness and sampling cost.
> >
> > 4. Because the original description of Contribution 3 (Line 98) causes some misunderstanding for reviewer QhYu, we revise this part to provide a more accurate and precise explanation of the superiority of our method.
> >
> > 5. Because the original description of Contribution 1 (Line 94) led to misunderstanding for reviewer QhYu regarding the distinction between effectiveness and compatibility, we have revised the statement to avoid ambiguity.
> >
> > 6. In accordance with reviewer 5d7r's feedback, we add the wall-clock time curves and the grad_norm & GSS curves to Appendix A6.3 and A6.4.
> >
> > 7. We have added a discussion of VinePPO to the related work section.
> >
> > 8. We have updated the analysis of variance in Section 4.2.3 to further clarify the theoretical contributions of our paper.
> > 9. **We have included our complete training code and the trajectory prediction dataset in the supplementary material to facilitate reproducibility verification.**
> >
> > Regarding the additional experiments—including comparisons across more model sizes and architectures and additional baselines—these require more substantial LaTex formatting adjustments and are not yet fully integrated into the main paper. We plan to include an additional section to present these experiments comprehensively.
> >
> >
> > We conclude by sincerely thanking all reviewers and the Area Chair for the care and insight you have invested in evaluating our submission. Your detailed feedback and the time you devoted—especially under the pressures of the rebuttal season—are truly appreciated.

---

### Author Response · Authors · 2025-11-27
**Public Response on Paper Update**

We sincerely thank all reviewers for their thoughtful and constructive feedback. Following your suggestions, we have prepared and uploaded an updated version of the paper. To facilitate your review, all revisions are clearly highlighted in **blue**.

In addition to the nine updates previously summarized—covering improvements to the introduction, the reformulated GRPO/GRPO-MA presentation, clarifications of Contributions 1 and 3, refinement of algorithm descriptions, strengthened theoretical analysis, added diagnostic curves in the appendix, and the inclusion of code and datasets—**we have now also integrated the remaining experimental results**. This newly added section incorporates the more baselines (Sec 5.3), model-size comparisons (Sec 5.4), and architectural analyses (Appendix A6.5).

We are grateful for the reviewers' guidance, which has substantially improved the clarity and completeness of our manuscript. We hope the updated version alleviate any remaining concerns and provides a more coherent and comprehensive presentation of our contributions.

---

### Author Response · Authors · 2025-11-29
**Final Response [1/2]**

We sincerely thank the new Area Chair for evaluating our submission under the unexpected circumstances caused by the recent OpenReview incident. We also appreciate the original Area Chair and the reviewers for their careful reading and constructive feedback. We understand the additional workload created by the situation and are grateful for the opportunity to provide this final response. Our goal here is to summarize how our rebuttal engages with and clarifies the reviewers' original questions, and we hope this helps the new Area Chair in their assessment.

The reviewers highlighted several strengths of our submission:

1. Reviewer Q536 and 5d7r appreciated the clear writing, motivation, and articulation of GRPO’s limitations.
2.  Reviewer JCTn and 5d7r emphasized the theoretical grounding, noting that the variance analysis via the delta method provides a meaningful connection between multi-answer sampling and variance reduction.
3.  Reviewer QhYu and Q536 considered GRPO-MA a simple and effective improvement strategy, with multi-answer sampling offering more stable reward signals.
4. Reviewers Q536, JCTn, and QhYu also recognized the breadth of our experimental evaluation across mathematics, code, visual reasoning, and manipulation, and Reviewer 5d7r pointed to the lower Gradient Spike Score as quantitative evidence of improved training stability.


We also note the four common concerns raised across the initial reviews, all of which we addressed in **detail in the Common Response and the reviewer-specific Official Comment**. Below we briefly summarize each concern and how we resolved it:

1. **Missing stronger baselines (DAPO, GRPO-CARE, Dr.GRPO):** We added these three methods as additional baselines and reported new results on the trajectory prediction task.
2. **Validation on larger models and different architectures:** We conducted further experiments using Qwen2.5-VL-7B-Instruct as a larger model and Qwen3-4B-Instruct as a different architecture, again on the trajectory prediction task.
3. **Concerns about novelty and contribution:** We clarified our methodological contributions and will further **elaborate on them in this final response [2/2].**
4. **Whether a single thought can produce multiple different answers:** We explained that high-temperature sampling is a standard setting in GRPO-style methods and naturally induces diverse outcomes in LLMs. We also provided statistical evidence demonstrating that a single thought indeed yields multiple distinct answers in practice.

The other concerns raised by the reviewers have also been clarified in our point-by-point official responses.

### We additionally included our training code and the trajectory prediction dataset in the supplementary material to support reproducibility.

---

> ### Author Response · Authors · 2025-11-29
> **Final Response [2/2]**
>
> To further clarify our contributions and novelty, we summarize them here.
>
> ### **1. A unified perspective that motivates a simple and effective modification.**
> We revisit GRPO-style reasoning RL from a unified perspective that brings together three interconnected issues: (i) gradient coupling between thoughts and answers, (ii) the sampling–reward trade-off under sparse rewards, (iii) and instability in advantage estimation. Viewing these challenges collectively reveals a common underlying mechanism, which in turn leads to a simple yet effective modification of the sampling procedure: generating multiple answers for each thought. **Under this unified view, the multi-answer strategy is not an ad hoc tweak, but a natural and principled adjustment that directly alleviates the shared root of these challenges.**
>
> ### **2. Novel theoretical analysis using the multivariate delta method.**
> To the best of our knowledge, we are the first to analyze variance in advantage estimation for GRPO-style algorithms using the multivariate delta method and to derive an approximate closed-form expression. The analysis highlights two key properties: (i) increasing **the number of thoughts (K)** causes the variance to plateau at a non-zero constant, and (ii) increasing **the number of answers per thought (M)** drives the variance toward zero. These findings show that increasing K alone cannot eliminate variance, while increasing M provides a necessary and effective way to obtain more stable and accurate advantage estimates. This new variance characterization forms one of our core contributions to the community.
>
> ###  **3. Broad and heterogeneous empirical validation.**
>
> We evaluate our method across one of the widest task ranges studied in GRPO-style works, including text reasoning (math, code), multimodal tasks (object detection, trajectory prediction, affordance prediction, demand prediction, OCR-based VQA), and object manipulation tasks. In response to reviewer suggestions, we further added experiments on a larger model (Qwen2.5-VL-7B-Instruct) and a different architecture (Qwen3-4B-Instruct). These results collectively demonstrate that **the method is robust across domains, modalities, model sizes, and architectures**. To the best of our knowledge, such broad and heterogeneous validation has not previously been conducted for GRPO-type improvements.
>
> ### **Summary**
>
> Overall, although the method is intentionally simple, its simplicity does not imply limited novelty or contribution. Our contributions to the community lie in providing (i) a simple, effective, and general improvement method for GRPO-style reasoning RL, (ii) a new variance analysis that clarifies when and why multi-answer sampling is necessary for stable advantage estimation, and (iii) extensive empirical validation across multiple domains, model sizes, and architectures. This combination positions GRPO-MA not merely as a practical tweak, but as a principled and broadly applicable paradigm for improving GRPO-style reinforcement learning.

---

### Meta-Review · Area_Chair_FmJs · 2025-12-10

**Summary:**

Reviewers were initially unanimous to reject the paper, but one reviewer (5d7r) agreed to raise their score during the rebuttal.

The common concern was the lack of comparison to recent, stronger baselines (DAPO, GRPO-CARE, Dr.GRPO). The other concerns were specific technical or mathematical details unique to each reviewer.

**Reviewer Concerns:**

Lack of comparison to stronger baselines
- The authors added DAPO, GRPO-CARE, Dr.GRPO to their experiments, and one reviewer already raised their score in response. I think this concern has been well-addressed.

Specific technical or mathematical concerns
- I believe some of these concerns might become less important in light of the additional experiments, but overall they remain unaddressed.

Novelty
- One reviewer was concerned that the method is too similar to GRPO. The authors pointed out that they included novel theoretical analysis, but I note that this does not directly address the reviewer's concern about the method's originality. I think some merit can be awarded to the theoretical analysis.

**Reviewer Scores:**

5d7r: already agreed to increase their score

Q536, JCTn, QhYu: I think 1 out of 3 would have raised their score, on account of the new baselines

I think the overall scores would end up tied. I appreciate the authors' efforts to address reviewers concerns. However, LLM reasoning is a highly competitive topic this year, and there were many stronger papers than this one to recommend acceptance for.

---

### Decision · Program_Chairs · 2026-01-26

Reject